Manuscript prepared for Geosci. Model Dev.
with version 4.2 of the LaTeX class copernicus.cls.
Date: 11 September 2017

# Shingle 2.0: generalising self-consistent and automated domain discretisation for multi-scale geophysical models[†]

Adam S. Candy[1] and Julie D. Pietrzak[1]

[1]Environmental Fluid Mechanics Section, Faculty of Civil Engineering and Geosciences,
Delft University of Technology, The Netherlands

*Correspondence to:* Adam S. Candy (a.s.candy@tudelft.nl)

[†]Library code, verification tests and examples available in the repository at
https://github.com/shingleproject/Shingle. Further details of the project presented at
https://www.shingleproject.org.

**Abstract.** The approaches taken to describe and develop spatial discretisations of the domains required for geophysical simulation models are commonly ad hoc, model or application specific and under-documented. This is particularly acute for simulation models that are flexible in their use of multi-scale, anisotropic, fully unstructured meshes where a relatively large number of heterogeneous

parameters are required to constrain their full description. As a consequence, it can be difficult to reproduce simulations, ensure a provenance in model data handling and initialisation, and a challenge to conduct model intercomparisons rigorously.

This paper takes a novel approach to spatial discretisation, considering it much like a numerical simulation model problem of its own. It introduces a generalised, extensible, self-documenting ap-

10 proach to carefully describe, and necessarily fully, the constraints over the heterogeneous parameter space that determine how a domain is spatially discretised. This additionally provides a method to accurately record these constraints, using high-level natural language based abstractions, that enables full accounts of provenance, sharing and distribution. Together with this description, a generalised consistent approach to unstructured mesh generation for geophysical models is developed, that is au-

15 tomated, robust and repeatable, quick-to-draft, rigorously verified and consistent to the source data throughout. This interprets the description above to execute a self-consistent spatial discretisation process, which is automatically validated to expected discrete characteristics and metrics.

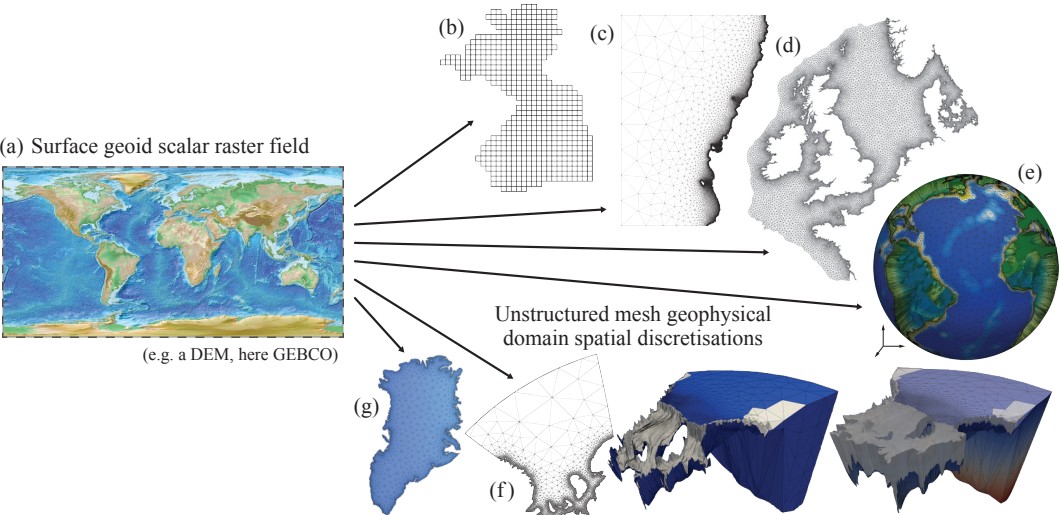

Fig. 1: The challenge: to generate a self-consistent domain discretisation approach for geophysical domains that is generalised such that it can be applied to a wide range of applications, with new domains efficiently prototyped and iterated on, and is fully described such that the process can be automated, is reproducible and easily shared. (a) shows a typical source Digital Elevation Map (DEM) dataset (that naturally lend themselves to structured grid generation) used to produce a regular grid of the Atlantic Ocean (e.g. under a format-native land mask) in (b), and a selection of unstructured mesh spatial discretisations: (c) Bounded by part of the Chilean coastline and a meridian. (d) North Sea. (e) Global oceans. (f) Grounding line of the Filchner-Ronne ice shelf ocean cavity up to the 65°S parallel, with surface geoid mesh $\mathcal{T}_h$, full mesh $\mathcal{T}$ with ice-ocean melt interface highlighted, and accompanied by ice sheet full discretisation. (g) Greenland ice sheet.

## 1 Introduction

Numerical simulation models have become a vital tool for scientists studying geophysical processes.
Mature operational models inform continuously updated short-term public weather forecasts, whilst studies of mantle dynamics and ice sheet evolution improve understanding of physical systems in relatively inaccessible locations, where data is sparse.

Use of unstructured mesh spatial discretisations[1] is growing in the fields of modelling geophysical systems, where it is possible to conform accurately to complex, fractal-like surfaces and vary
spatial resolution to optimally capture the physical process, or multi-scale range of processes under study. The past few years have seen a global unstructured ocean model (FESOM, Sidorenko et al., 2014) join structured studies in internationally coordinated climate studies, such the Coupled Model Intercomparison Project (CMIP, Meehl et al., 2007; Taylor et al., 2012) and the Coordinated Ocean-ice Reference Experiments (CORE Griffies et al., 2014, and accompanying studies in the
*Ocean Modelling* special issue). More are in active development (e.g. Ringler et al., 2013) and the number of unstructured models joining these efforts – that directly contribute to reports compiled by the Intergovernmental Panel on Climate Change (IPCC) – is likely to grow. Similarly, on smaller

---

[1]For the purposes of the discussion here, *spatial discretisation* specifically refers to the division of a continuous spatial domain into discrete parts — a discrete tessellation or honeycomb — a generalised notion of triangulation.

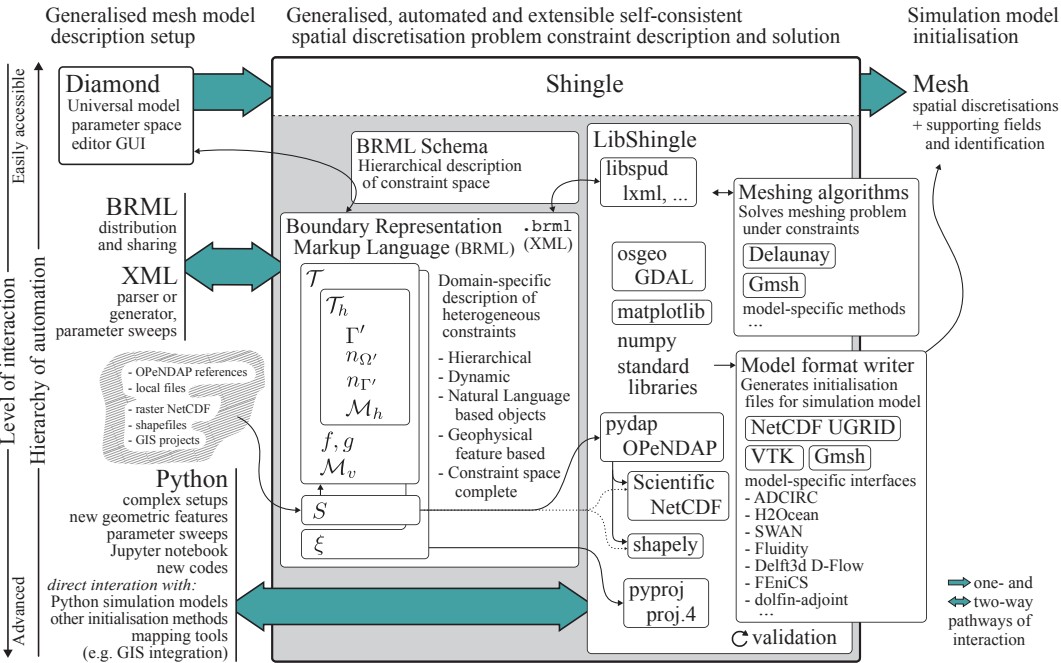

Fig. 2: A schematic illustrating the generalised approach to flexible unstructured mesh specification and generation for geophysical models. The hierarchy of automation (tenet 7) is highlighted, from a relatively simple high-level interaction: Diamond GUI ↔ Shingle → Mesh, to complex low-level development communicating with the LibShingle library. Nomenclature defined in section 2.

scales, the geometric flexibility of unstructured discretisations are being applied to reduce the need for nesting models, and in accurately applying forcings or coupling physics (e.g. Kimura et al., 2013)

on complex and possibly dynamic, deformable physical interfaces. At the cusp where these efforts meet, prospects for introducing successively greater complexity in the representation of coastal seas in global ocean models are reviewed in Holt et al. (2017).

The challenge (see figure 1) of constraining and fully describing an arbitrarily unstructured spatial discretisation bounded by complex, fractal-like bounds that typically characterise geophysical

domains, with inhomogeneous and potentially anisotropic spatial resolution, is a significant one. Defining the domain geoid bounds is no longer a simple case of applying a land mask to similarly regular gridded data. The generalised constraints are now a heterogeneous set of functions (Candy, 2016), and as a consequence are more difficult to describe. In general, domain discretisations are often under-described leaving it difficult to repeat simulations exactly, which particularly for the

unstructured case, can have a strong influence on model output. Not only is the description and generation process a significant challenge, but achieving this in a way that maintains a record of provenance such that simulations as a whole are reproducible, that scales and is efficient, and consistent to source data – attributes required and expected in scientific modelling studies – make this a much more difficult problem (summarised in table 1). Existing, standard structured-mesh tools

cannot be used.

Grid generation for geophysical models in real domains is not only becoming a significantly more complex and challenging problem to constrain and describe, but additionally in the computational processing required. As models include a greater range of spatial scales, more computational effort is required to optimise the discretisation before a simulation proceeds (e.g. the actively developed

MPAS models, Ringler et al. (2013), strongly optimise their hexagonal prism based mesh discretisation). An increasing number of geometric degrees of freedom demand the meshing process is broken up over multiple parallel threads (as demonstrated in Candy, 2016), just as simulation models have evolved to run in parallel.

---

1. Accurate description and **representation of arbitrary and complex boundaries** such that they are contour-following to a degree prescribed by the metric size field, with aligned faces so forcing data is consistently applied ($\Gamma'$, $f$, $g$).

2. **Spatial mesh resolution** to minimise error; with efficient aggregation of contributing factors, ease of prototyping and experimentation of metric functions and contributing fields, over the entire extent of the bounded domain ($\mathcal{M}_h$, $\mathcal{M}_v$).

3. Accurate geometric **specification of regions** and **boundary features**; to provide for appropriate interfacing of regions of differing physics, model coupling and parameterisation application ($n_{\Omega'}$, $n_{\Gamma'}$).

4. **Self-consistent**, such that all contributing source data undergoes the same pre-processing, ensuring self-consistency is inherited.

5. **Efficient drafting and prototyping** tools, such that user time can be focused on high-level development of the physics and initialisation of the modelled system.

6. **Scalability**, with operation on both small and large datasets, facilitating the easy manipulation and process integration, independent of data size.

7. **Hierarchy of automation**, such that individual automated elements of the workflow can be brought down to a lower-level for finer-scale adjustments.

8. **Provenance** to ensure the full workflow from initialisation to simulation and verification diagnostics are reproducible.

9. **Standardisation of interaction** to enable interoperability between both tools and scientists.

---

Table 1: The *nine tenets of geophysical mesh generation*, that solutions to the spatial discretization of geophysical model domains should address (from Candy, 2016).

These challenges are identified in Candy (2016) by the *nine tenets of geophysical mesh generation*,

summarised in table 1. This work takes the view that significant progress can be made towards these by approaching the mesh generation problem in the same way as a numerical simulation model.

Simulation domains in geophysical models are typically defined with reference to geographical features. A tsunami simulation geoid surface domain is, for example, usually described by a length of coastline between two points (commonly marked by longitude or latitude references) extended

out to an orthodrome. In order to demonstrate the method, a worked example of the 2010 Chile tsunami is presented, starting here and concluding with figure 7. This follows the relatively simple

high-level interaction: Diamond GUI to Shingle to mesh, that is illustrated across the top of figure 2. In this case, with the earthquake centred about 35.9°S 72.7°W (see figure 7), the domain is concisely described:

> " ... *bounded by the 0m depth coastline from 32°S to 40°S,*
>
> *extended along parallels to the 77°W meridian,* (∗)
>
> *in a latitude-longitude WGS84 projection...*"

As part of the generalisation of domain description, this new approach interacts directly with these natural language based geographic references, structured by a formal grammar, to provide a general, model-independent and accurate description of spatial discretisation for geophysical model domains. This forms part of the Shingle (2011–2017) computational research software library, that accompanies this work, providing a novel approach to describing and generating highly multi-scale boundary-conforming domain discretisations, for seamless concurrent simulation.

The objective of this paper is to provide:

1. A user-friendly, accessible and extensible framework for model-independent geophysical domain mesh generation.

2. An intuitive, hierarchical formal grammar to fully describe and share the full heterogeneous set of constraints for the spatial discretisation of geophysical model domains.

3. Natural language basis for describing geophysical domain features.

4. Self-consistent, scalable, automated and efficient mesh prototyping.

5. Platform for iterative development that is repeatable, reproducible with a provenance history of generation.

With significant progress made through the novel approach of considering the problem much like that of a numerical simulation model problem.

The previous work Candy (2016) developed a consistent approach to domain discretisation, with a focus on uniform processing and data sources, which further enabled the discretisation of domains not possible with standard approaches. Additionally, it identified the complete set of heterogeneous constraints required to fully describe a mesh generation problem for the discretisation of geophysical domains. This work now extends and generalises this consistent approach introducing a natural language based formal grammar for a modeller to describe and share the constraints. Under the formal grammar the description is ensured necessarily complete, such that the problem is fully constrained and is therefore reproducible. This employs the novel hierarchical problem descriptor framework Spud (Ham et al., 2009) which has been specifically designed to manage large and diverse option trees for numerical models. The formal self-describing data file is a universal, shareable description of the full constraints, written in a standard data format, presented in context through a natural hierarchical structure, readable by established open source libraries.

The pathways of interaction with the library have grown (outlined in figure 2), such that it is accessible to a wide range of users. It has a modular library framework, with for example, geospatial

operations, homeomorphic projections, meshing algorithms and model format writers the focus of distinct modular parts. This together with the use of standard external libraries where possible allows development to remain in small sections of the code base such that developers can stay within their specialisms. Additionally, the dictionary approach to managing option parameters taken by Spud means new features can be added and exposed through interfaces, such as the Diamond Graphical User Interface (GUI), without the need to pass new arguments through code functions, and similarly require small changes and only in low-level code.

Output writers in the library prepare the solution discretisation for use in simulation codes, in cases where the output Python objects cannot be used directly, encouraging the use of standard formats and also supporting existing proprietary model-specific formats. These additionally support supplementing the spatial discretisation (which itself includes a vector field describing mesh node coordinate locations) with additional interpolated fields for simulation model initialisation and forcing (figure 2).

Through both the objects in the problem description file (figures 3 – 5) and those in the Python library LibShingle (figure 6), Shingle provides a language to combine geographic components to build up boundary representation, mesh spatial variation and identification – a high-level abstraction to the complex constraint description problem – which is then processed by the library in deterministic (or as close to as possible) process to accurately construct the specified mesh in a repeatable way.

The validation tests of Candy (2016) have been significantly widened from the limited boundary representation tests to include expected discrete properties and metrics of the high fidelity description and resulting domain discretisation. These expected characteristics are prescribed as part of the self-describing problem file, such that other users can check the output is as intended. This self-contained description and validation is then straight-forwardly processed by the library verification engine, making it easy to add new tests.

Through this approach, geophysical domain discretisation can be the relatively simple steps (top of figure 2) of using the Diamond GUI to choose a dataset and specify bounds using natural language objects, which is then run through the Shingle executable to produce a mesh. This is accessible and straightforward to new users. More so with the suite of test cases that provide examples and easily ensure verification through a built-in test engine.

More advanced use can be built up in stages through the GUI, with validation checks on expected mesh properties easily added to ensure reliable reproduction throughout the iterative mesh prototyping process. Beyond this the description is based on Extensible Markup Language (XML) which is easily interrogated and modified with standard tools. Lower-level still, the natural language based objects and discretisation constraints can be accessed directly through its Python library interface. This has grown since its first iteration reported in Candy (2016), where it was used to develop complex discretisations dependent on the mean position of Antarctic Circumpolar Current (ACC) and

domains to complex grounding line positions under the floating ice shelves of Antarctica. Python plugins for QGIS (Quantum GIS Development Team, 2016) were developed using parts of the Shingle library code to demonstrate integration with Geographic Information Systems (GIS) in Candy et al. (2014).

With mesh generation becoming a complex problem to describe and a computationally challeng-
ing process, that we argue is best handled in an approach that mirrors the development of a numerical simulation model, support and interaction with other frameworks such as GIS is best maintained with a standalone library and a formal problem description specifically designed to constrain the general geophysical domain discretisation problem.

The paper is structured such that the following section 2 sets out the challenge, reviewing the set
of heterogeneous constraints $1-5$ required to fully describe a domain discretisation problem, and key considerations in table 1. The natural language based BRML problem description is introduced in section 3, with a consideration of source data in section 4. The LibShingle library, central to the generalised approach (illustrated in figure 2), is detailed in section 5 and ways to interact with the framework are presented in section 6. Examples and validation are covered in section 7, with
conclusions made in section 8.

## 2   Generalised unstructured spatial discretisation for geophysical models

### 2.1   Constraints for mesh generation in geophysical domains

The contrast in dominant dynamical processes that characterise geophysical systems, split in orthogonal directions parallel and perpendicular to the local gravitational acceleration $g$, leads to a
spatial decoupling that restricts the parameter space of general spatial domains $\Omega \in \mathbb{R}^3$. Meshes of geophysical domains can be built differently in these distinct directions in order to well-support the associated dynamics, with mesh characteristics on the geoid plane considered independently of those in the perpendicular direction of $g$. A formal description of the heterogeneous set of constraint functions, homeomorphic mappings and topological spaces, required to fully describe geophysical model
domain spatial discretisations, is developed and detailed in Candy (2016), of which a summary of the key outcomes follows.

**Constraints:** *The spatial domain discretisation for a computational geophysics simulation in a domain $\Omega \subset \mathbb{R}^3$, requires the constraint of*

1. ***Geoid boundary representation*** *$\Gamma_g$, of the geoid surface $\Omega_g \subset \mathbb{R}^3$, inclusive of the maximal extent of $\Omega$ perpendicular to $g$. Under a homeomorphic projection $\xi$, this is considered as the chart $\Omega' \subset \mathbb{R}^2$, such that the boundary $\Gamma'$ is described by*

$$\Gamma' : t \in \mathbb{R} \mapsto \zeta(t) \in \mathbb{R}^2, \tag{1}$$

*an orientated vector path of the encompassing surface geoid bound defined in two-dimensional*

 *parameter space.*

2. ***Geoid element edge-length resolution metric*** *for dynamics aligned locally to a geoid, described by the functional*

$$\mathcal{M}_h : \boldsymbol{x} \in \Omega' \mapsto \mathcal{M}_h(\boldsymbol{x}) \in \mathbb{R}^2 \times \mathbb{R}^2. \tag{2}$$

3. ***Boundary and region identification***, *prescribed by*

$$n_{\Gamma'} : t \in \mathbb{R} \mapsto n_{\Gamma'}(t) \in \mathbb{Z}, \textit{ and} \tag{3}$$
$$n_{\Omega'} : \boldsymbol{x} \in \Omega' \mapsto n_{\Omega'}(\boldsymbol{x}) \in \mathbb{Z}, \textit{ respectively.} \tag{4}$$

4. ***Surface bounds***, *height maps defined on the surface geoid domain, described by the functions*

$$f, g : \boldsymbol{x} \mapsto \mathbb{R} \quad \forall \boldsymbol{x} \in \Omega'. \tag{5}$$

5. ***Vertical element edge-length resolution metric*** *for dynamics in the direction of gravitational acceleration (e.g. buoyancy driven), described by the functional*

$$\mathcal{M}_v : \boldsymbol{x} \in \Omega \mapsto \mathcal{M}_v(\boldsymbol{x}) \in \mathbb{R}. \tag{6}$$

## 2.2 Decoupled mesh development

The spatial decoupling permits discretisation in two stages corresponding to directions parallel and perpendicular to the local gravitational acceleration (refer to figure 3). Firstly, the 'horizontal' geoid surface domain discretisation problem is solved under constraints $1-3$ using the surface geoid boundary representation $\Gamma'$ (1), geoid element edge-length metric $\mathcal{M}_h$ (2), with boundary and region identifications, $n_{\Gamma'}$ (3) and $n_{\Omega'}$ (4) respectively, such that

$$h : \{\Gamma', \mathcal{M}_h, n_{\Gamma'}, n_{\Omega'}\} \mapsto \mathcal{T}_h, \tag{7}$$

a tessellation of $\Omega' \subset \mathbb{R}^2$, with identification elements.

Secondly, if needed, this is followed by discretisation in a direction aligned with gravitational acceleration. The constraints 4 and 5, describing the surface bounds $f$ and $g$ (5) and vertical edge-length metric $\mathcal{M}_v$ (6), together with the surface geoid discretisation $\mathcal{T}_h$ (7), forms a discretisation problem that is solved through the process

$$v : \{\mathcal{T}_h, f, g, \mathcal{M}_v\} \mapsto \mathcal{T}, \tag{8}$$

to give the full domain discretisation of $\Omega \subset \mathbb{R}^3$, consisting of a tessellation or honeycomb together with identification of the boundary and internal regions (i.e. $n_{\Gamma'}$ and $n_{\Omega'}$).

## 2.3 The nine tenets of geophysical mesh generation

Accompanying the constraints, Candy (2016) identifies the nine attributes listed in table 1 as key to geophysical mesh generation processes.

## 2.4 Tessellation algorithms and existing grid generation approaches

The algorithms to form the required unstructured tessellations are an ongoing, active area of research. In the above these are required for $h$ and $v$ in (7) and (8) to produce $\mathcal{T}_h$ and $\mathcal{T}$ respectively. Delaunay triangulation (Delaunay, 1934), originating over eighty years ago, is now well-established and the basis for the majority of methods applied in flexible mesh finite volume and finite element numerical simulations.

The general-purpose three-dimensional meshing library Gmsh (Geuzaine and Remacle, 2009) has been used to make significant progress in ocean modelling on unstructured meshes (e.g. see Legrand et al., 2000; White et al., 2008; van Scheltinga et al., 2010; Gourgue et al., 2013; Thomas et al., 2014). In their application to modelling geophysical systems, developments further constrain mesh structure for specific geophysical features or numerical discretisation requirements, such as grid orthogonality. As well as providing easy access to established, robust algorithms such as the anisotropic Delaunay method of George and Borouchaki (1998) and frontal algorithm of Rebay (1993), treatments for specific features in Earth system domains have been added (see Lambrechts et al., 2008) to enable successful geophysics simulations on this solid base. In van Scheltinga et al. (2010), the nearest neighbour algorithm of Arya et al. (1998) applied in Gmsh is modified to improve representation of narrow channels in the highly irregular oceanic archipelagos of the Arctic. For model requirements, the spring-based force equilibrium approach of Conroy et al. (2012) regularise meshes for finite volume C-grid discretisations of shallow water equation models which depend on orthogonal grids. Orthogonal grids are further optimised in Holleman et al. (2013) to construct grids aligned to dominant flows. Additionally, algorithms are being adapted to the evolution in computational resources with Jacobsen et al. (2013) presenting algorithms enabling the construction of grids in parallel based on a Delaunay approach.

Whilst the development of tessellation algorithms has improved spatial discretisation it is difficult to develop, describe and share the complex constraints required. Vector illustration packages and the Scalable Vector Graphics (SVG) data type have been used to develop the orientated vector path $\Gamma'$ together with Gmsh for the tessellation operation $h$ in Gourgue et al. (2009), de Brye (2011) and Kärnä et al. (2011). This has been shown to successfully enable the inclusion of complex bounds in model simulations, but relies on hand-editing and in a non-geospatial environment that does not natively preserve measures of space or consider projection mappings. A plugin (Legrand

et al., 2007; Lambrechts et al., 2008) for Gmsh successfully includes sections of the pre-prepared coastline contours of the Global Self-consistent, Hierarchical, High-resolution Geography (GSHHG, Wessel and Smith, 1996). This also has been shown to work well in limited applications, restricted to pre-prepared bounds. It is also possible inconsistencies can This also has been shown to work well, but is restricted to pre-prepared bounds and limited applications. Candy et al. (2014) demonstrated the use of GIS to develop constraints in a geospatially-consistent approach. This too is an accurate solution that takes into account geospatial measures, but in itself does not resolve all of the tenets, since it may not be possible for large cases or those with many complex features which require an automated treatment.

Successful, high quality spatial discretisations are dependent on both these tessellation algorithms faithfully operating to prescribed constraints and that the set of constraints are a consistent and accurate representation of the model domain. The focus of this work is not to improve the numerical tessellation algorithms, which have been successfully employed in geophysical applications. We leave this part to well-established libraries such as Gmsh, building upon these with interaction through standardised Application Programming Interfaces (APIs) and data structures. We focus on the latter, on the constraints, following the five objectives of this paper listed in section 1, to develop a generalised, model-independent approach to their creation. This is extensible to accommodate future demands in model development and application studies. Unlike previous approaches that have tended to be bespoke with specific, limited application, the aim here is to work towards the 'holy grail' of the nine key considerations or 'tenets' listed in table 1.

## 3 Boundary Representation Markup Language

### 3.1 Unstructured domain discretisation — a model problem

The functional forms (1)–(6) of the unstructured meshing problem require a range of types of data, from more standard two-dimensional raster maps, to tensors and orientated vector paths. It is a challenge to manage this heterogeneous collection of parameters (tenets 5 and 8), such that they are handled consistently (tenet 4) and for the level of complexity that can be encountered (tenets 6 and 7). This is in contrast with the structured mesh case, which requires relatively simple data of the same format as its inputs: a two-dimensional Digital Elevation Map (DEM) raster dataset supplying a two-dimensional raster mask, for example.

Mesh specification in the unstructured case, with flexibility to include conforming boundaries, is much more like the initialisation of a numerical simulation model. This typically contains a heterogeneous set of functions: those defined over $\mathbb{R}^3$ initialising or forcing full fields, together with boundary conditions defined on surfaces in $\mathbb{R}^2$ and potentially line and point sources, or full field functions of reduced rank such as the gravitational acceleration parameter, or value of a bulk eddy viscosity, for example. Mesh descriptions and constraints are only going to become more complex

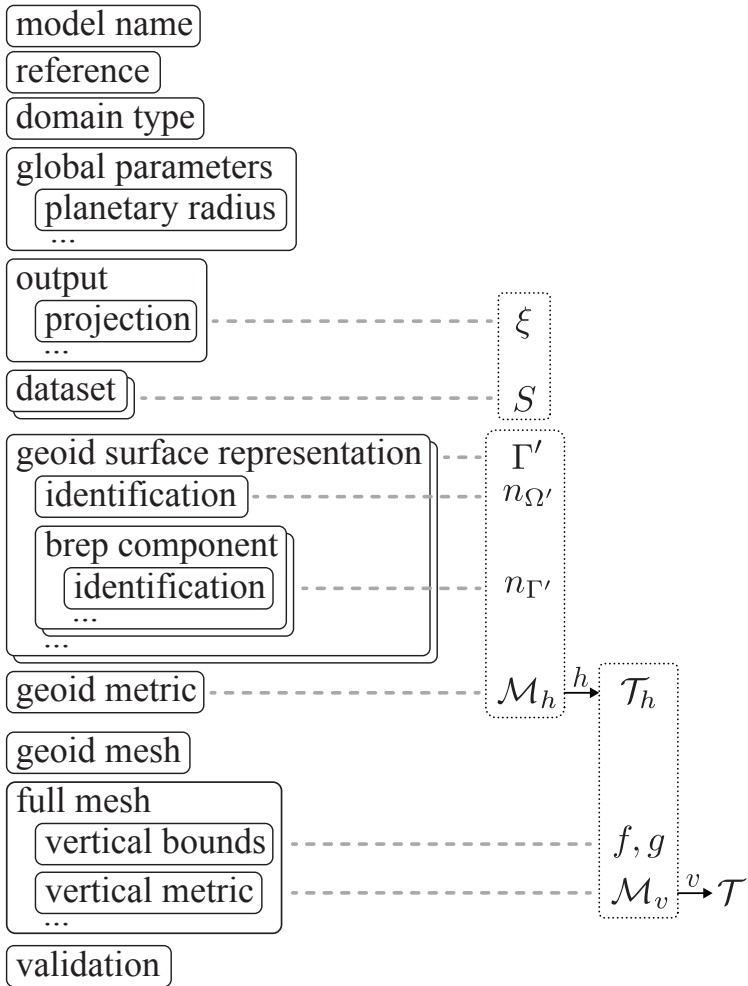

Fig. 3: Overview layout of geophysical domain mesh constraint description highlighting extensible dynamic components and correspondence to source data $S$, projection $\xi$ and constraints $1-5$.

as simulation models include a larger range of spatial scales and physical processes. Moreover, like a simulation model, unstructured mesh generation includes calculations that can be computationally

demanding. The generation of conforming boundary representations is no longer a simple binary operation identifying which elements lie in the simulation domain through mask fields. Similarly, the construction of domain discretisations with variable element sizes contains many more unknowns in the unstructured case than the corresponding local cell-division approaches typically used to increase spatial resolution in the structured case.

In light of this, Shingle takes the approach that domain discretisation specification and generation is best considered as a model problem. Formalised, the output mesh is the solution of a discretisation problem under a heterogeneous parameter space of constraints.

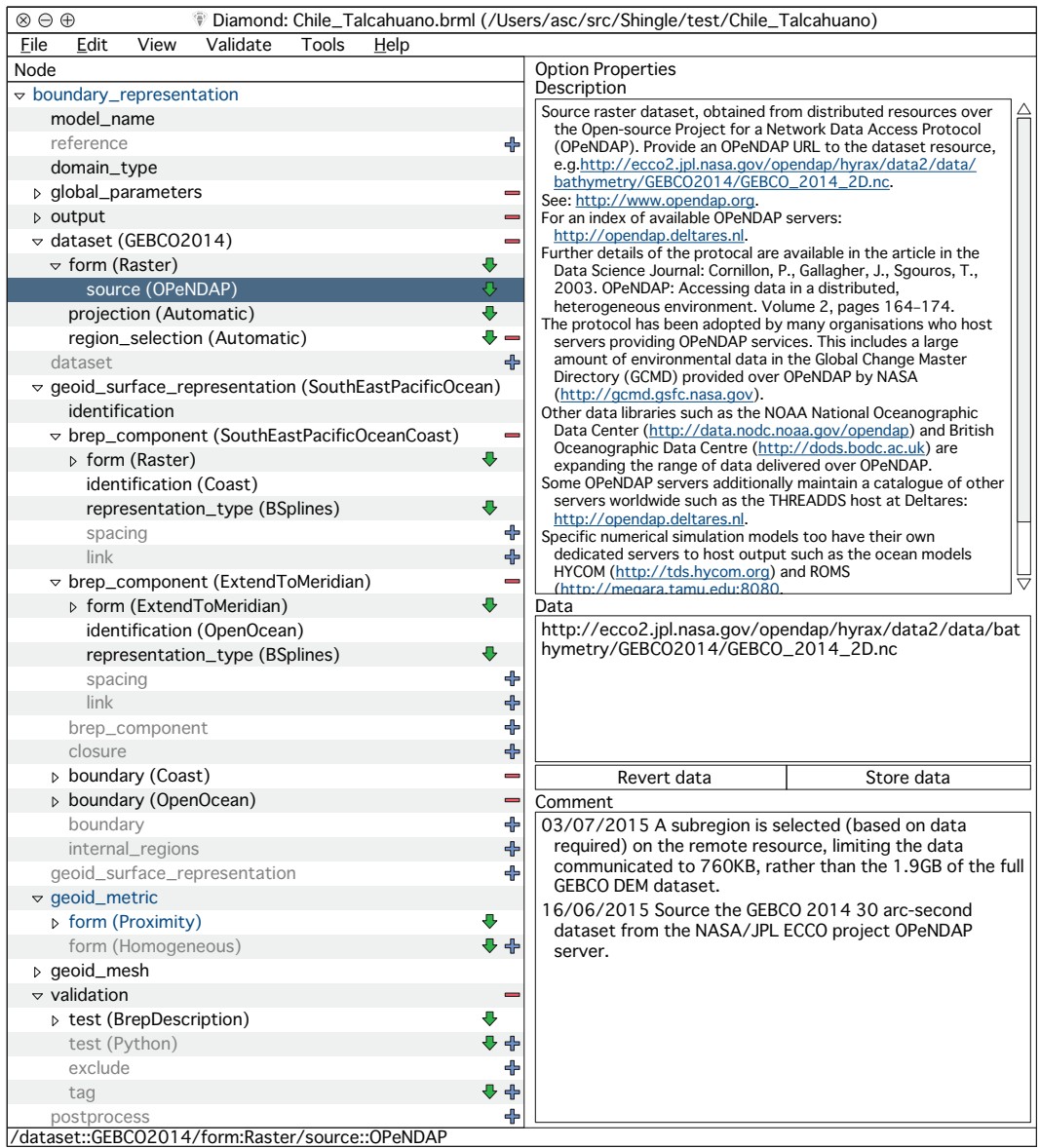

Fig. 4: An example view of the Graphical User Interface Diamond inspecting the hierarchical tree of option parameters that fully constrain the geophysical domain mesh problem. Each node is shown in context on the left, with their option properties presented on the right, including raw data and the possibility to note comments. This is guided by the BRML schema developed and supplied with Shingle, which additionally provides the fuller self-describing option descriptions shown in the top right. Options down the tree highlighted in blue are mandatory and guide the user to defining a complete set of constraints.

## 3.2 Spud constraint space management

Much like numerical model input parameter specification, mesh generation is often overlooked, and

a secondary consideration to the dynamical core of a numerical model. Typically inputs are ad hoc,

```xml
<?xml version='1.0' encoding='utf-8'?>
<boundary_representation>
  <model_name>
    <string_value lines="1">Chile_Talcahuano</string_value>
  </model_name>
  <global_parameters/>
  <output>
    <projection>
      <string_value>LatLongWGS84</string_value>
    </projection>
  </output>
  <dataset name="GEBCO2014">
    <form name="Raster">
      <source name="OPeNDAP" file_name="http://ecco2.jpl.nasa.gov/
      ↪   opendap/hyrax/data2/data/bathymetry/GEBCO2014/ GEBCO_2014_2D.nc"/>
    </form>
    <projection name="Native"/>
    <region_selection name="Automatic"/>
  </dataset>
  <geoid_surface_representation name="SouthEastPacificOcean">
    <identification>
      <integer_value rank="0">9</integer_value>
    </identification>
    <brep_component name="SouthEastPacificOceanCoast">
      <form name="Raster">
        <source name="GEBCO2014"/>
        <region>
          <longitude>
            <minimum>-77.0</minimum>
            <maximum>-71.0</maximum>
          </longitude>
          <latitude>
            <minimum>-40.0</minimum>
            <maximum>-32.0</maximum>
          </latitude>
        </region>
        <contourtype name="coastline0m"/>
        <comment>Simple single bounding box centred about the epicentre 35.909S
        ↪   72.733W.</comment>
      </form>
      <identification name="Coast"/>
      <representation_type name="BSplines"/>
    </brep_component>
    <brep_component name="OpenMeridian">
      <form name="ExtendToMeridian">
        <longitude>
          <real_value rank="0">-77.0</real_value>
        </longitude>
      </form>
      <identification name="OpenOcean"/>
      <representation_type name="BSplines"/>
    </brep_component>
    <boundary name="Coast">
      <identification_number>
        <integer_value rank="0">3</integer_value>
      </identification_number>
    </boundary>
    <boundary name="OpenOcean">
      <identification_number>
        <integer_value rank="0">4</integer_value>
      </identification_number>
    </boundary>
  </geoid_surface_representation>
  <geoid_metric>
    ...
  </geoid_metric>
  <validation>
    <test name="BrepDescription" file_name="data/Chile_Talcahuano.geo.bz2">
      <compressed/></test>
    <test name="NodeNumber"> ... </test>
  </validation>
</boundary_representation>
```

Fig. 5: Example domain discretisation specification, in a self-describing BRML file (with a few parts marked ... skipped). This is a human-readable translation of the simple description (∗) under the formal grammar of the schema that defines the geophysical domain discretisation constraint space. This file is examined by the GUI in figure 4 and, on straight-forward and automated processing by Shingle, produces the simulation-ready spatial discretisation of figure 7.

model-specific, plain text files containing name lists that are expanded as a model develops. For only but simple cases, this leads to model interfaces (and their associated pre- and post-processing tools) that are difficult to maintain and simulation setups that are not easily shared and understood.

This problem of model input parameter specification is considered in Ham et al. (2009), together with the proposed solution Spud. This provides a generalised, model-independent method of describing all constraints to a model problem, that is dynamic, easily extensible with a hierarchical context for parameters. Formal grammars guide user input, minimise errors and formalise parameter specification.

### 3.3 Constraint space description

The options available to describe a mesh discretisation are typically defined by model interfaces. These tend to be ad hoc and unportable, tied directly to numerical simulation codes. Initialisation tools then require their own implementation to interpret and write model options, which is prone to error and potential inconsistencies.

Existing file formats have been used, and their syntax overloaded, to describe geophysical spatial discretisations. Ice sheet domains are built up using a Constructive Solid Geometry (CSG) approach within the COMSOL (COMSOL; Li et al., 2009) multi-physics modelling environment in Humbert et al. (2009). The GeoCUBIT (Casarotti et al., 2008) branch of CUBIT developed for seismic inversion domains, and a plugin for Gmsh (Geuzaine and Remacle, 2009) to enable the creation of domains bounded by paths from the GSHHG (Wessel and Smith, 1996) database. Extensions to GIS (e.g. Candy et al., 2014) enable a flexible development of geoid surface boundary representations. Extensibility of these frameworks for the purposes of geophysical domain discretisation and model initialisation is limited, with for example GIS frameworks being built up from working on two-dimensional raster fields. Similarly, project files associated with GIS do not contain all of the information required to fully constrain a spatial discretisation problem, and moreover, it is not possible to include the high-level natural language functional descriptions proposed here. As Candy et al. (2014) demonstrates though, GIS methods can benefit geophysical domain development, and their role is included in the schematic figure 2.

Use of Spud enables a description of model option parameter space to be considered separately. This is constructed in a schema file, a machine readable specification of which options are expected, their type and context, and how they should be read: A formal grammar to be used to describe model constraints. The constraints $1-5$ that fully describe the geophysical domain discretisation problem have been structured into a schema. A schematic of the included components and their relationship to required constraints is shown in figure 3.

This is a single hierarchical and formal description of the constraint space, and more generally the options available to the user in generating a mesh. As illustrated in figure 2, it is part of Shingle and is central to how components of the approach interact with BRML files that describe a particular

meshing problem. At the simplest, highest level use of Shingle, this is transparent to the user. For more advanced use and development, it provides a centralised and language-based description of the constraint space that all other parts of Shingle, and the geophysical mesh generation process,

depend.

### 3.4   Dynamic, hierarchical parameter description

Just like the case of a numerical model, there are a wide range of possible options in mesh generation, even when restricted to geophysical problems. The BRML schema builds on the general schema language for simulation models prepared in Ham et al. (2009), to give an option-complete language

for the mesh generation problem. This is exactly the type of purpose Spud is intended for and other current models in development are adopting this approach to formally describe model constraint spaces, like for example the new TerraFERMA model of Wilson et al. (2016).

   This caters for options which may be specified multiple times, at potentially varying levels of option hierarchy in multiple contexts. For example, as the block diagram of figure 3 highlights, a

simulation domain can contain multiple geoid surfaces $\Gamma'$, each with potentially multiple boundary representation components (e.g. simple orientated polylines with identification). BRML is an XML language, and by nature is hierarchical and extensible. With this structure, and guided by what the schema permits (itself representing the constraints $1 - 5$), it is easy to dynamically add, repeat, expand and remove options and groups of options whilst in context.

As an example, use of the Spud framework immediately provides access to the Diamond GUI which enables easy editing and drafting of new domain discretisations. This GUI uses the schema file (see figure 2) to guide navigation of the option tree. Through this the GUI knows to expect at least one definition of a geoid surface $\Gamma'$, for example, and a specification of a geoid metric $\mathcal{M}_h$ (and requires these from the user). Additional geoid surfaces or more feature-rich boundary

representation components are easily added and built up at a later stage, dynamically increasing the complexity of the mesh generation problem.

### 3.5   Option tree cross-references

Options are structured into a hierarchical tree within the BRML description. The grouping of constraints $1 - 5$ and decoupling (section 2.2) are naturally structured in this way, as figure 3 highlights.

This is much like numerical simulation model options parameters, which motivated the development of Spud and adoption of an underlying XML based language.

   In some cases there exist dependencies across the option tree, and these are achieved through attribute names. For instance, the choice has been made to centralise source dataset definitions. These are named (e.g. 'GEBCO2014' in figures 4 and 5) and this name referred back to whenever the

data is required. This is also used to assign potentially multiple boundary representation component sections to the same named boundary identification (e.g. the 'Coast' and 'OpenOcean' named

identifications of figures 4 and 5).

This also allows component boundary representations sections to be used multiple times. This is required, for example, when distinct physical regions meet at an interface (e.g. the open ocean meets an ice sheet) and share a boundary. The component boundary representation section defining the interface can then be referred to out of the order defined by the hierarchy, and from potentially separate parent geoid surface representation $\Gamma'$ (where for instance $\Gamma'_o$ and $\Gamma'_i$ are setup to represent neighbouring geoid surface representations for the ocean and ice, respectively).

### 3.6 Natural language descriptions

Domains for geophysical simulations are typically described with reference to bounding lines on orthodromes such as meridians and parallels, together with global or segments of contours such as a 0m coastline, for example. More generally, geographic features are identified with a similar combination. The Southern Ocean for example, is defined extending up from the Antarctic coast to the $60°$S parallel, and the Atlantic and Indian Oceans divided at the $20°$E meridian.

This is the natural way to identify bounds for geophysical models. Setting up these geographic bounds and including all features contained within in a format suitable for meshing algorithms can be a time consuming, difficult to edit and repeat, ad hoc process. Shingle automates this and from a basis of natural language definitions typically used in geophysical modelling studies.

The original consistent boundary representation generation approach described in Candy (2016) enabled sections of contours to be selected and domains extended meridionally to parallels. This has been generalised significantly to allow a wide range of arbitrary bounds described with natural language definitions. Moreover these can be defined multiple times, and in context with hierarchy available within the BRML description. In the example presented in figures 4 and 5 the boundary representation can be seen to include two components: a section of the Chilean coastline and a second extending the domain out to a meridian at $7°$W, mirroring those in the description ($*$).

### 3.7 Arbitrary and discrete descriptions

More flexible functional descriptions can be made within the BRML written directly in Python. This again in a relatively readable form, using primitives such as the positions 'longitude' and 'latitude', or Universal Transverse Mercator (UTM) coordinates 'x' and 'y'. This can be used to describe an arbitrary orthodrome, for example.

In addition to this, the natural language basis can be supplemented with raw discrete data types such as orientated polylines from the GSHHG database, mapping databases (e.g. the UK national Ordnance Survey OpenData resource) or those developed directly in a GIS as Candy et al. (2014) demonstrates, bounding a domain to the complex UK coastline together with the fine man-made structures of Portland Harbour. The high fidelity boundary representation is not only built up from components constructed on-the-fly from functional forms referencing geographic features, but also

discretised forms containing an explicit description of domain constraints, if needed (see figure 2).
These are available through the central dataset section of the option hierarchy (figure 3), and accessed
from local or distributed resources.

### 3.8 Self-describing constraint options

The constraint space description developed in the BRML schema is self-describing, containing a
verbose description of each option. This information can be presented alongside options in the GUI
(see the top right of figure 4, for example) or reported for any option errors occurring at run time,
again from this centralised constraint space descriptor resource, the schema. In this way the schema,
and as a result the GUI, act as a manual, directly supporting users as mesh options are made.

From the developer's perspective, this Spud based approach means new features can be added
with minimal code changes. The XML based structure means codes focus on patterns of options.
The schema defines what expected and the code loops through the hierarchy following well-defined
patterns, picking up options from a corresponding in-memory dictionary tree.

For the user, mesh generation with real fractal-like boundaries can be as simple as selecting a
coastline segment by a bounding box and on the other side a bounding orthodrome, with choice of
element edge-length metric (see figure 7).

### 3.9 Provenance record

A complete description of the domain discretisation problem is a fundamental requirement if an ac-
curate record of provenance is to be made, and this is provided by the BRML file. These BRML files
alone are themselves easily parsable XML based problem description files, human-readable with
structure. This is focused on a textual natural language problem description and is lightweight as a
result such that changes are easily tracked with version control systems such as Git and Subversion
(SVN).

Together with the problem description, the BRML maintains details of authors responsible for
their creation, contact details, comments including timestamped notes on past changes made in de-
velopment (seen in figures 3 and 4). This is similar to the record kept within the global attribute
metadata contained in NetCDF headers, which is supplemented through operations performed on
the data with tools such as the Geospatial Data Abstraction Library (GDAL). The ADCIRC hydro-
dynamic circulation model (Westerink et al., 2008) makes a record of this type of information in
its NetCDF output, inherited from its initialisation namelist files. Shingle records this information
in output where possible, notably the high fidelity boundary representation, supplementing it with a
record of the library release version and unique repository abbreviated commit hash. Unique iden-
tifiers of other libraries are also recorded, such as the version of the meshing tool employed (e.g.
Gmsh).

## 4 Source data management

Data contributing to discrete domain characterisations can be large in size, difficult to distribute efficiently and computationally costly to process. The current version of the global bathymetry dataset GEBCO (2014) containing only elevation is currently 1.9GB in size, for example. Efforts are growing to provide a complete provenance record of numerical model simulations, with direct instructions from research funders requiring a research data management plan (NWO Data Management Protocol, 2014) and in general, accountability from the public, it is important to detail data source origin and content accurately.

Options for the management of mesh generation source data range from:

1. Recast data into form suitable for distribution and share with BRML description.
2. Distribute processed datasets with BRML irrespective of size.
3. Begin from a standardised raw dataset, and conduct potentially computationally demanding processing as needed.
4. Refer to remote repositories of source data, such that data is downloaded and processed on demand.

Often this data processing stage of the mesh generation process is not well-described, and difficult to reproduce, with filtering, subsampling and agglomeration operations only loosely outlined.

Modern data descriptors support a record of provenance (such as the 'history' field embedded in NetCDF, Rew et al.), so it would be possible to record the filtering, subsampling and other processing here or within the BRML.

The purpose of the BRML description of constraints is to provide an accurate description of the meshing problem. It is not the intent to reinvent new standards for data description. Along this line of design, with a focus on provenance record and how data is handled, and noting the computational demands and connectivity speeds that affect options 3 and 4 above will continue to improve in the future, the approach is made to depend directly on raw, standard and potentially remote data sources.

### 4.1 OPeNDAP integration

The problem of efficient access to large remotely hosted data sources is tackled by Cornillon et al. (2003) which describes OPeNDAP (Open-source Project for a Network Data Access Protocol). The protocol has since been adopted by many organisations who host servers providing OPeNDAP services. This includes a large amount of environmental data in the Global Change Master Directory (GCMD) provided over OPeNDAP by NASA[2]. Other data libraries such as the NOAA National Oceanographic Data Center[3] and British Oceanographic Data Centre[4] are expanding the range of

---

[2]http://gcmd.gsfc.nasa.gov
[3]http://data.nodc.noaa.gov/opendap
[4]http://dods.bodc.ac.uk

data delivered over OPeNDAP. Some OPeNDAP servers additionally maintain a catalogue of other servers worldwide such as the THREADDS host at Deltares[5]. Specific numerical simulation models too have their own dedicated servers to host output such as the ocean models HYCOM[6] and ROMS[7].

This has typically been applied to sharing geophysical model output data in combination with the (NetCDF Rew et al.) and Climate and Forecast (CF, Gregory, 2003) metadata standards (Hankin et al., 2010), for intercomparisons and post-processing analysis. Here we apply OPeNDAP to model initialisation. In Shingle, this OPeNDAP negotiation is achieved using the standard Python library pydap. In this way Shingle can request fundamental operations are applied to distributed datasets before they are delivered for further processing, picking out required fields and regions of interest to reduce the size of data communicated. A description of further processing such as subsampling and filtering is then maintained in the BRML and executed through standardised Python wrappers to established geospatial tools such as GDAL. A reference in place for the GEBCO (2014) data source hosted on the NASA/JPL ECCO OPeNDAP server is made in figure 5, where the region of interest (for cropping on the remote server) is automatically established by its use further down in the tree.

Keeping the BRML focused on problem description, with references to source data, ensures it is lightweight and portable. Iterative adjustments to the mesh generation are also then made with changes to descriptions rather than data. Furthermore, these are then easily managed in version control systems.

This additionally ensures the verification test engine is lightweight and apart from a dependence on standard software libraries, and a connection to OPeNDAP servers, is self-sufficient and can be easily be setup and used independently.

Constraints built from distributed resources are encouraged, but to engage with existing mesh generation workflows and as a pragmatic solution, source files can be cached or local files used directly (see figure 2).

### 4.2 Self-consistent boundary representation development

Shingle applies the self-consistent approach to mesh generation developed in Candy (2016). Within the BRML description this is emphasized through a central data source definition (seen in figures 3 – 5), rather than external sources brought in directly at different levels in the hierarchy and correspondingly in the generation process (figure 3). It is then easier to ensure datasets and their component fields undergo the same pre-processing to generate high fidelity constraints that are consistent, and a solution spatial discretisation that is self-consistent.

Data used to construct the spatial domain discretisation is commonly a DEM describing a surface through perturbations from a reference geoid surface (e.g. to establish a geoid surface boundary representation), but is not limited to this form, with for example Candy (2016) developing a mesh

---

[5]http://opendap.deltares.nl
[6]http://tds.hycom.org
[7]http://megara.tamu.edu:8080, http://tds.marine.rutgers.edu

optimised to the mean track of the ACC, based on currents in the Southern Ocean.

## 5 LibShingle, the Shingle library framework

### 5.1 Built on standard libraries

The library LibShingle is written in Python and uses standard libraries for operations where possible (see table 2). It can simply be used transparently through the Shingle executable to interpret the constraints specified in BRML file descriptions. For lower level more advanced use building up constraints for more complex setups or in prototyping natural language objects for automating the inclusion of new geographic features, interaction can be made directly with the LibShingle library
as figure 2 illustrates.

| Function | Dependency |
| --- | --- |
| Problem specification (complete, extensible BRML description) | – Spud library: multipurpose, model-independent, parameter management based on formal grammars for general simulation models. |
| Visual interfaces and GUIs | – Diamond: An automated graphical user interface linked to Spud that guides valid model input choices. |
| | – Geographic Information Systems (GIS) for geospatially aware vector pathline and DEM processing (e.g. QGIS). |
| Tessellation algorithms | – Gmsh library, including: anisotropic Delaunay (George and Borouchaki, 1998), the frontal algorithm of Rebay (1993) and local modification technique of Lambrechts et al. (2008). |
| | – Triangle Library Python Bindings (2014). |
| Efficient large dataset handling | – Geospatial Data Abstraction Library, GDAL. |
| | – Standard Python libraries, including Numpy, ScientificPython and Scientific.IO. |
| Geometrical operations | – Standard Python shapefile library, shapely. |
| | – Standard Python data analysis and plotting library, matplotlib. |
| Geospatial operations (e.g. projection-aware area calculations) | – proj.4 through the Python library, pyproj. |
| | – GDAL (as above). |
| | – QGIS (as above). |
| Data sourcing and remote access | – OPeNDAP through the Python library, Pydap, |

Table 2: Function of external dependencies and libraries applied in the approach.

Mirroring the BRML constraint description (overviewed in figure 3), the library contains natural

language based objects that can be built up in code to construct components of a mesh generation problem, including boundary representations and element edge length metrics. The mesh problem can then be solved under these constructed constraints all within a Python context.

LibShingle uses the open source Python shapely library (refer to figure 2) to handle polyline imports and manipulations. The Scientific.IO library is relied on to efficiently process raster NetCDF files. The homeomorphic projections to the charts required in the mesh generation process (see Candy, 2016), such as $\xi$ of (1) are interpreted and managed by the Proj.4 Python library pyproj. Geospatial operations can be made by both high-level Shingle objects, or built up with GDAL oper-

ations through its Python osgeo interface.

(a)

```python
from shingle import SpatialDiscretisation, Dataset, Boundary
# Set up constraints
R = SpatialDiscretisation(name='NorthSea')
R.SetProjection('UTM', -3, 52) # alternatively zone='30U'
gebco = Dataset(type='raster', source='opendap', url='...', region=[-12,14,45,62])
coast = Boundary('coast', id=3)
S = R.AddSurface()
S.AddBoundaryComponent(source=gebco, contour='ocean0m', id=coast)
...
M = R.Discretisation()
11 M.Save('NorthSea.msh')
```

(b)

```python
# Modify boundary representation output projection
import pyproj
p = pyproj.Proj('+proj=utm +zone=30U +ellps=WGS84 +datum=WGS84 +units=m")
R.SetProjection(p)
R.Save('NorthSea_UTM30U.brml')
```

(c)

```python
# Simple parameter sweep example
from shingle import Load
R = Load('Weddell_Sea.brml')
S = R.GetSurface('SouthernOcean')
B = S.GetBoundaryComponent('OpenParallel')
for latitude in [float(x) for x in xrange(-75,-65,2)]:
B.ExtendToParallel(latitude)
B.Save('Weddell_Sea_and_%.0f.brml' % latitude)
```

Fig. 6: Example interactions with the Shingle Python library LibShingle. (a) Using natural language constructs native to Shingle, counterparts to BRML entries. (b) Together with objects native to external libraries. (c) Loading, extending and saving descriptions from BRML.

Although the use of external libraries may require updates to Shingle in the future to maintain compatibility, this is minimal compared to the benefits of using standardised implementations (tenet 9), that have community effort to ensure ongoing support with operating systems and interaction with other software and methods.

**5.2  Low-level interaction through Python objects**

In addition to the ongoing support from standard libraries in high-level use, Shingle has been written to interact directly with external libraries. Objects such as pyproj projections, GDAL operations,

surface and polyline descriptions can be used interchangeably with LibShingle. An example bringing in a UTM projection setup externally using the standard library pyproj is shown in figure 6(b). This supplements the high-level text-based natural language definitions available in the BRML, and a route to adding new high-level boundary representation BRML objects to LibShingle as needed.

### 5.3 Efficient parameter space exploration

In developing a new application study applying a numerical simulation model, it is common to iterate on a spatial discretisation until it is optimum and fit for purpose. This involves small changes in the constraints, exploring parameter space often through a loose bisecting binary search algorithm. This process can be rigorously implemented and automated with LibShingle, where modifications are guided by the schema describing the formal grammar of the constraint space through libspud. Figure 6(c) illustrates a simple template to modifying and generating a range of BRML mesh descriptions. The solution mesh discretised domains can be generated in the same way, and this could further be used to initiate numerical simulation runs.

This algorithmic formulation of constraints is easily extended to enable complex operations that are difficult to achieve with other approaches. For example, the loop of figure 6(c) is trivially extended to include a search algorithm exploring a parameter space to converge a domain discretisation on a required total number of nodes and hence degrees of freedom.

Being an XML based language, the BRML descriptions can also be simply interrogated and modified directly with standard XML libraries. This interaction is highlighted separately in figure 2.

### 6 Model, method and data interaction and interoperability

Shingle has been built with modules for high-level interactions, with established tools applied in mesh generation. These are highlighted in figure 2 and, where possible, interaction is achieved through standardised Python APIs, such as the Triangle Library Python Bindings (2014) for the Triangle (Shewchuk, 2002) library of Delaunay mesh algorithms. It is important that the approach is not limited to a particular meshing approach or library, in line with the ninth tenet on standardisation. In the applications presented here and those routinely tested by the verification engine, algorithms within the Gmsh library are used for the tessellation process $h$ of (7). It is applied because, as described in section 2.4 it does a very good job at adhering to the constraints provided, is relatively robust and provides access to multiple tessellation algorithms through a common API, again in support of tenet 9. In this case a high fidelity boundary representation is output in Gmsh format using a specific format writer developed within a collection of writer modules that are a part of Shingle.

Similarly, fields supporting a meshed domain (e.g. initial full-field temperature state) can be output as unstructured VTK files, using a format writer extending standard VTK libraries. Data

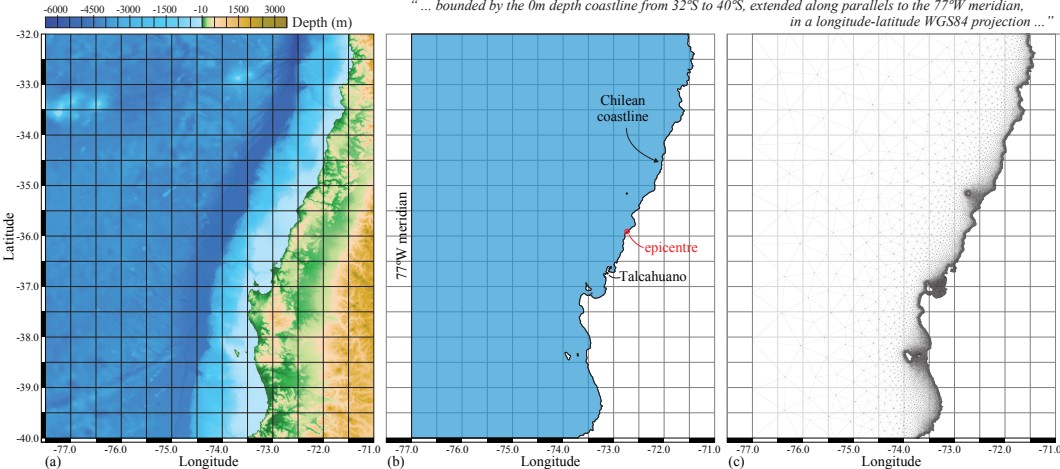

Fig. 7: Example simulation domain for modelling ocean wave propagation and tsunami inundation in the 2010 Chile M8.8 earthquake, centred at 35.9°S 72.7°W, approximately 100km north of Talcahuano. This domain is relatively simply described by (∗) in section 1 with constraints formally defined by the BRML of figure 5 (with some further description and corresponding formal BRML to constrain spatial resolution). Generation is a simple matter of translating the former into the latter under the formal grammar, with both being human-readable descriptions. Shingle automatically handles the details of defining a high fidelity boundary representation $\Gamma'$ in (b) from the GEBCO (2014) DEM (a) and, notably here, includes island features to give a geoid surface representation with non-zero genus (following the approach of Candy, 2016), and further to automatically produce a simulation-ready meshed spatial discretisation $\mathcal{T}_h$ in (c).

is written and stored efficiently in an XML based data format containing blocks of binary data compressed using the zlib library.

### 6.1 Model format writers

Models with non-standard data formats are supported through specific format writers. This modular approach enables new format writers (and readers) to be added as needed. As examples, Shingle includes modules to prepare initialisation files for the ADCIRC hydrodynamic circulation model and H2Ocean shallow water equation model.

   As well as writing mesh solutions, the output writers are used for validation purposes and in the
general purpose efficient prototyping (tenet 5). Output can be prepared for viewing alongside source data in geospatially valid context provided by GIS frameworks, with for example the resulting mesh and discrete bounds overlaid over DEMs directly within GIS (see Candy et al., 2014). This is useful for a visual evaluation of conformity, to see how well geographic features are represented. For large discretisations, visualisations tools designed specifically for efficiently handling large unstructured
datasets can be employed, such as Paraview[8], which is directly supported by Shingle using VTK.

   Interaction at different levels is important to ensure a hierarchy of automation tenet 7. Particu-

---

[8]http://www.paraview.org

larly challenging meshing problems can, for example, easily be offloaded to more capable dedicated resources. For quick visual inspection purposes, Shingle can automatically output an image of the geoid surface mesh discretisation.

## 6.2 Input readers

Parallel to the writer modules, Shingle includes readers. These are used to interact with meshing libraries where needed, loading in output mesh discretisations produced by Gmsh on-the-fly, for example. Additionally this can be used to support a wider range of data sources and initialisation. Standard data in NetCDF and shapefile form can be read. Readers here can import more complex heterogeneous data, including GIS projects with multiple layers containing a wide range of data types, for example.

## 6.3 Embedding in model codes

As a Python library unifying boundary representation constraint and solution, LibShingle makes it possible to incorporate complex domain discretisation of real geophysical domains in overarching model control scripts, which is where development of new cutting-edge models is headed (see for example, Rathgeber et al., 2015; Pelupessy et al., 2016). In this way the model supplements the problem constraints sent to LibShingle (see figure 10), dependent on numerical discretisations employed in the simulation model, and the BRML would be truly independent of specific models, a pure description of the boundary representation, resolution and identification. Moreover, interaction through the library enables models to handle the output discretisation directly as the Python objects constructed by Shingle, rather than an intermediate file object.

As Pelupessy et al. (2016) demonstrates, complex multi-model Earth system models can be created and coupled, and interactively monitored, on potentially a heterogeneous array of computational resources, all coordinated from a central a Python interface. LibShingle brings domain discretisation in real geometries to these type of extensible Earth system modelling frameworks.

## 7 Verification and discretisation validation

A suite of verification tests are provided together with Shingle, along with the automated test engine detailed in section 7.2. A selection of geophysical domain discretisations described in BRML that form part of the test examples are shown in figures 1 and 7 – 9. Each test is evaluated using validation tests built into Shingle and their BRML descriptions, as outlined in section 7.1. The test engine can be used to verify a new install, and flexibly to support iterative mesh drafting and prototyping (tenet 5).

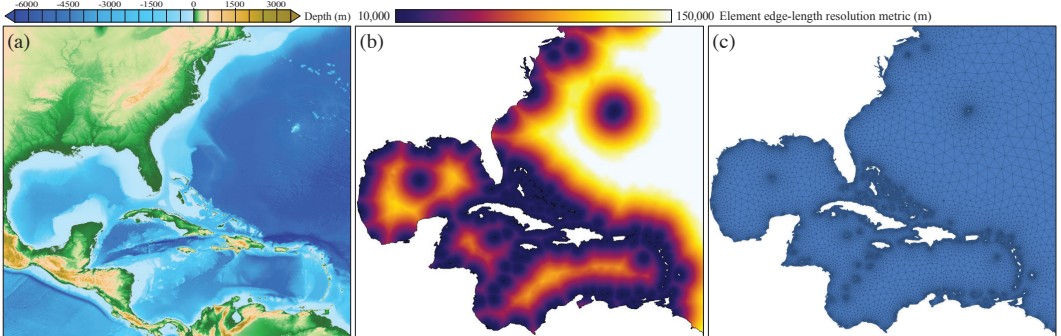

Fig. 8: Simulation domain focused on the Caribbean Sea basin. (a) GEBCO (2014) DEM. (b) Surface geoid element edge-length resolution metric $\mathcal{M}_h$ developed as a function of (a). (c) Surface geoid boundary representation $\Gamma'$ in blue, overlaid with multi-scale spatial discretisation $\mathcal{T}_h$.

### 7.1 Self-validation

Validation of the mesh generation process is achieved in four ways. Firstly, with reference to the formal grammar of the constraint space, a degree of self-validation can take place on-the-fly as mesh options are built up. Following rules described in the schema, only some options are available and certain combinations permitted. Unlike with namelist descriptions, or ad hoc collections of data, the user does not need to wait until running Shingle before receiving feedback on option validity. Available options are limited dynamically following the constraints and option selections. Moreover, with information from the schema on the mesh generation problem, it is possible to identify which options are required for the problem to be complete. The creation of a new BRML file immediately requires a name, type and options to be completed for at least one geoid surface representation and a geoid metric. The GUI highlights which required options remain to be completed (see figure 4). This is particularly useful to users new to mesh generation.

Secondly, the required 'type' option classifies the mesh and checks at runtime it is suitable for the intended simulation. A 'shallow water' model requires only a surface geoid discretisation $\mathcal{T}_h$ for example, whilst a full three-dimensional mesh is needed in other simulation types. This is a sanity check to ensure the mesh generation problem is fully constrained for the intended purpose, beyond the fundamental constraints $1 - 5$.

Thirdly, a parsing stage following application of a meshing algorithm eliminates commonly found issues in output mesh descriptions, ensuring structural integrity. For example, additional lone, un-connected boundary elements are removed in this step to ensure the discretised output mesh is as expected. Meshing algorithms do not usually possess information on underlying numerical dis-cretisations, and it is also possible elements are generated that 'tied' to boundary conditions, with no independent free unknowns. This type of problem in the spatial discretisation is often difficult to identify, only being picked up at runtime, or through careful visual inspection. This parsing is an opportunity to identify and process these at this stage. Numerical simulation codes are some-

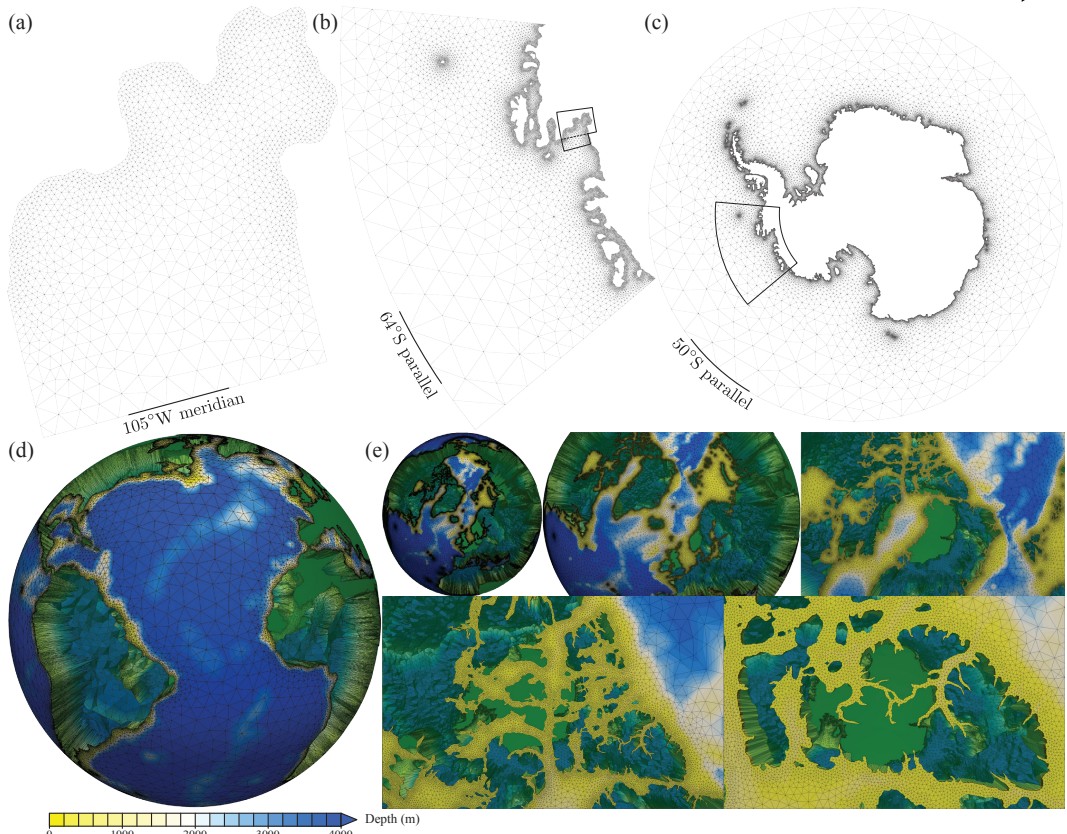

Fig. 9: A selection of further example geophysical domain discretisations straight-forwardly described in BRML and automatically constructed using Shingle. (a) $\mathcal{T}_h$ of the Pine Island Glacier ice shelf ocean cavity from ice-bedrock grounding line extended out to the 105°W meridian. (b) The Amundsen Sea region in West Antarctica extended out to the 64°S parallel. (c) The Southern Ocean Antarctic continent landmasses, from ice grounding line to 50°S parallel, built from a high fidelity boundary representation containing 348 automatically identified islands. (d) The full $\mathcal{T}$ of the global oceans, with a radial scaling of 300 to exaggerate the vertical extent of the discretised shell and land regions shaded green. (e) Zoomed in regions focusing on the complex Canadian Arctic Archipelago west of Greenland around Ellesmere and Baffin island. (a)–(c) are generated from the GEBCO (2014) DEM and presented under a orthographic projection centred on 90°S, and (d)–(e) from RTopo (Timmermann et al., 2010) and viewed in a Cartesian frame. These contain a multi-scale of spatial resolutions, with element edge-lengths parallel to the geoid in these examples, specified through $\mathcal{M}_h$, ranging from 2km to 500km. Vertical layers in (d), specified through $\mathcal{M}_v$, vary from 2m to 500m, under differing regimes in a generalised hybrid coordinate system described further in Candy (2016), and leads to a mesh containing 8,778,728 elements and 35,114,912 spatial degrees of freedom under its discontinuous Galerkin finite element discretisation. Along with other examples presented in figure 1(c)-(g), these are part of the test suite accompanying the library.

times accompanied with standalone mesh checking tools to support initialisation stages (e.g. the MechChecker.F90 utility for the ADCIRC model), and visual interfaces can be used for manual inspection and editing, such as the Show Me tool provided alongside Triangle (Shewchuk, 2002) and the GUI of Gmsh (Geuzaine and Remacle, 2009). This is part of the mesh generation process and,

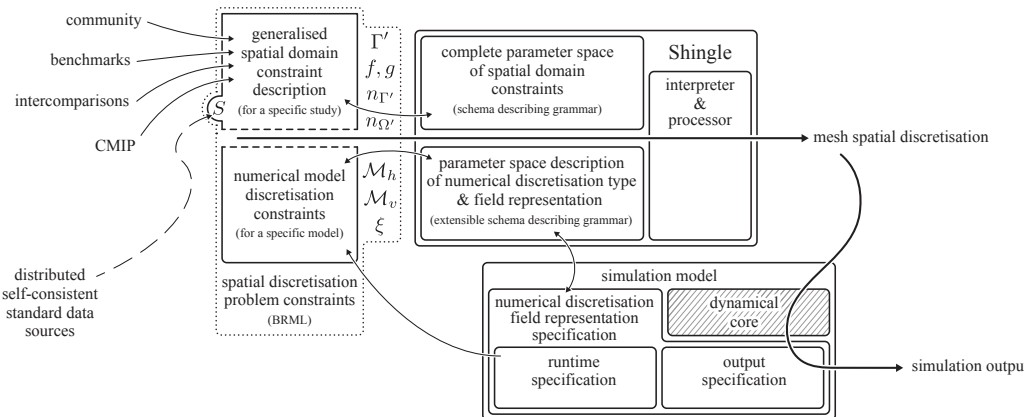

Fig. 10: Framework for generalised spatial domain discretisation for geophysical model simulations. A formal spatial domain constraint description (a model-independent grouping of high-level directives describing key geospatial boundaries and features, required spatial resolution and source datasets) for a specific study (e.g. the geography to include in a CMIP intercomparison study) is joined with specific constraints from a simulation model, depending on its internal numerical discretisations and field representations (e.g. following Gridspec (Balaji et al., 2007), or a UFL description (Alnæs et al., 2014)). These constraints are used by the interpreter Shingle to produce, in a robust, automated, repeatable process, a model-specific mesh spatial discretisation. Moreover, the latter description is further used to specify numerical simulation output representation (as CMIP uses Gridspec).

if possible, is better handled automatically following tenet 7, as proposed.

Lastly, the fourth approach to validation is through explicitly defined expected boundary representation and discretised mesh characteristics. Like the initial consistent approach of Candy (2016), the intermediate high fidelity boundary representation is compared at a raw level. Being a deterministic process, deviations are only expected as a result of the dependence on: Shingle library version and behaviour, linked libraries including tessellation algorithm implementations, source data and potential OPeNDAP response, machine precision and the originating BRML description. The potential for deviation in spatial discretisation output is considered in more detail in table 3, alongside approaches to mitigate these sources. At this stage of the meshing process, this has been supplemented with a test on the area within the bounds of the high fidelity geoid boundary representation $\Gamma'$.

On the discretised output, the tests include simple lower and upper bounds on output geoid mesh node and element numbers, the number of boundary elements, and element circumspheres to check adherence to metric constraints. The degree of representation is examined comparing the high fidelity geoid boundary representation surface area to its corresponding discretised form. Boundary complexity is measured through the overall Minkowski fractal dimension.

This provides a means for users to easily specify what should be expected in the discretised output, to ensure the accuracy required in tenets $1-3$. Testing built in to the mesh generation process, further automates the process. It is also important to ensure tenet 8: provenance, that the solution mesh is the same (within prescribed tolerances) as that that has been generated in the past, and potentially

| Dependency | Potential deviations and mitigation approaches |
|---|---|
| Source geophysical data | No changes are expected unless the resource, described by a unique Uniform Resource Locator (URL), itself is updated. Use of open digital archives, such as Zenodo[9], with resources linked to a Digital Object Identifier (DOI) ensure accessed data is identical, even for large geophysical datasets. |
| OPeNDAP server | Response should remain consistent. Server negotiation metadata recorded with output discretisation. |
| Tessellation algorithm | Prescribed by the BRML description. |
| Tessellation algorithm implementation | Tessellation libraries can change and lead to deviations. Library versions are recorded with the output discretisation and justifiable spatial tolerances specified in the BRML description. |
| Machine precision | Small influence and taken into account in verification tests. |
| Originating BRML description | BRML is problem complete. |
| Shingle model library | Library code will change. Test engine protects core functions and extends to validate key features of important cases. Library version number recorded in output. |
| Library dependencies e.g. Python, NumPy, ScientificPython, matplotlib, shapely, Scientific.IO, libspud, Pydap, pyproj, GDAL, QGIS, ... | Same situation as numerical simulation models using shared or statically-linked libraries. Where possible, routines are written to be robust to library changes. (e.g. library routine outputs further sorted by geospatial properties to ensure ordering invariant to linked library changes). |

Table 3: Dependencies and sources of potential deviation in the output spatial discretisations.

by others on different systems.

A self-validating description provides tenet 9: a standardisation of interaction with the descriptions themselves. Users can immediately begin building on and improving the work shared by others, having been able to check the descriptions give a solution expected by the creator. This eliminates ad hoc or purely qualitative measures of conformity and reinforces the provenance record of the mesh generation process.

This is important when these then form key components of critical studies, such as the coupled climate and Earth system models run for internationally coordinated model intercomparisons, such as CMIP and CORE (Meehl et al., 2007; Taylor et al., 2012; Griffies et al., 2014), that form the basis of reports compiled by the IPCC.

Models containing unstructured meshes with conforming boundaries are now starting to be used in such large-scale international research efforts (e.g. FESOM, Sidorenko et al., 2014). This approach provides the full provenance, reproducibility and complete constraining descriptions of the significantly more complex spatial discretisations supported by these models.

## 7.2 Continuous verification

To ensure Shingle as a whole continues to behave as expected for all users and on all systems, it contains a verification test engine. This processes a suite of key meshing problems, which are then automatically evaluated following the validation tests defined in their BRML description. Since the BRML descriptions are self-validating, the addition of new tests to the suite is simply a matter of adding the problem description file to a test folder of the source code. Testing is often a secondary consideration to new feature implementation, so it is important the extension of testing suite is as simple as possible.

This can simply be run at the time of a new installation, following the upgrade of required libraries or the operating system, or routinely as part of a commit-hook buildbot with dedicated resources to continuously verify new code pushed to a Shingle development code repository (see, for example, Farrell et al., 2011). Being built on standard libraries, it could further form part of an automated wider system framework validation, for the above climate intercomparison projects, for example, reproducing the entire process from initialisation to post-processing, on demand. Alternatively, the engine can be used to drive an efficient drafting and prototyping workflow (tenet 5) with updates to mesh generation problems automatically processed and tested, to support an iterative domain discretisation process.

## 8 Conclusions

This research has developed a high-level abstraction to mesh generation for domains containing complex, fractal-like coastlines that characterise those in numerical simulations of geophysical dynamics, together with a compact, shareable and necessarily complete description of the domain discretisation.

The approach is designed to be accessible to a wide range of users and applications. This begins at a simple standalone GUI-driven one way workflow, where users are guided through the option parameters required to constrain the domain discretisation problem. Options are presented in context through the hierarchical tree structure with documentation automatically provided alongside. Moreover, the use of a human readable XML format and introduction of high-level natural language based geographical objects give BRML problem constraint descriptions that closely follow those presented in literature and shared by scientists. The example built up from the description (∗), to BRML in figures 4 and 5, followed by the construction of the high fidelity boundary representation and resulting spatial discretisation shown in figure 7, highlights how the problem of generating a domain bounded by a complex coastline defined by a depth contour and three orthodromes, common in tsunami modelling studies, is trivially constructed and solved using Shingle.

This is easily built on and extended to larger and more complex problems. High-level objects automate processing of multiple, potentially complex geospatial features. BRML descriptions are

665 easily shared and XML sections cut and pasted to combine descriptions and build up complexity. New high-level objects and processing can be prototyped directly in Python to later join the core LibShingle operations library. Corresponding natural language based objects are available through the Python API, meaning domain discretisation can be achieved directly and purely in native Python code, for complex setups, direct integration with numerical simulation codes, or interactive sessions 670 or Jupyter notebooks[10]. Both the BRML file descriptor and modular LibShingle are extensible.

Extending the tsunami example shown in figure 7, this robust and automated approach could form part of a real time warning system using unstructured spatial discretisation, with a domain created on-the-fly, centred around the earthquake epicentre, in a direct response to measurement by GPS seismic monitors.

Recognising the domain discretisation process is becoming more challenging and more difficult to document completely, such that others can reproduce, has been central to steering this approach. Progress is focused on the nine tenets of geophysical mesh generation summarised in table 1. One result of this is that Shingle treats the mesh generation as a model problem. Strategies from numerical simulation model development have been adopted and modified to formalise the description 680 of the heterogeneous geophysical mesh generation constraints, such that they provide an accurate and complete description (tenets $1-3$) in a standardised language-based XML form (tenet 9). This compact text-based description easily affords a record of changes in the development of a domain discretisation (tenet 8) and through the BRML grammar ensures it is always a complete description and therefore reproducible. The model-based approach manages the range types of parameters 685 (which have diversified with the use of flexible unstructured discretisations) and supports users in their preparation, to allow for efficient drafting and prototyping (tenet 5). With options managed in a structured hierarchical tree, complex discretisations can be built up logically (tenet 7) and scaled up (tenet 6).

The creation of the BRML file boundary representation description is not intended to reinvent 690 standards. It is not a new data descriptor, for orientated vector paths or two-dimensional raster data, for example. There exist standards already that tackle these challenges well. It is rather a new problem descriptor, like those for Fluidity (Piggott et al., 2008) and the TerraFERMA model of Wilson et al. (2016), for fully describing the mesh generation problem specifically for geophysical model domains, following the approach that this requires solving the same types of challenges involved in 695 numerical model setup, that makes significant progress in meeting the tenets of table 1.

The consistent approach of Candy (2016) is adopted, with an emphasis on producing a self-consistent high fidelity description and resulting output domain discretisation. Consistency is additionally encouraged through a centralised definition of the source data and processing in the BRML description (see figures $3-5$). Use of decentralised, distributed datasets, efficiently accessed using 700 OPeNDAP, ensures the discretisation uses exactly the same source data on every processing instance.

---

[10]http://jupyter.org, examples available with the Shingle library source.

Verification and discretisation validation is achieved at multiple points throughout the process. The formal grammar of the BRML, imposed by the schema, enforces valid inputs and provides initial option checking. This framework and interaction with the schema using the libspud library additionally enables new self-validating user interfaces to be written. With expected mesh valida-
tion measures included in the BRML descriptions, discretisations are automatically validated and continuous verification of the library is easily obtained.

With the dependable, robustly verified library LibShingle for high-level abstractions for geophysical mesh generation, it is easily applied to develop interactions with other frameworks and models, such as GIS, as described in Candy et al. (2014). Critically, with the standalone LibShingle library, these are easier to maintain and better insulated to API changes in other codes.

It does not immediately solve the mesh generation constraint problem in general, since numerical simulation models use a wide range of mesh types and numerical discretisations. It has however, been designed with this in mind, with low-level structures that are extensible, to accommodate additional mesh types for example, and high-level constructs that are applicable to all geophysical models. Arguably the *'holy grail'* of domain initialisation for geophysical models, characterised by the constraints 1 – 5 following the development figure 3, is a grouping of high-level directives describing bounds (including key geospatial features to capture), required spatial resolution and source datasets that can be interpreted by any model, each dealing with the discretisation depending on the field representations within the model (figure 10). Shingle provides an extensible platform to achieve this, focusing on general, natural language based, model-independent descriptions of domain descriptions, that can be shared and used for different models. LibShingle additionally provides a means to interpret these descriptions such that this part of the process can be included in numerical simulation code, with the BRML constraints supplemented by those imposed by the simulation model, such as specific numerical discretisation choice (e.g. to use hexagonal over triangular prism elements), or ensuring a minimum degree of representation in maintained between bounds (e.g. within narrow river channel networks).

**Code availability, distribution and licensing**

The Shingle computational research software library, developed as part of this study, is available at https://github.com/shingleproject/Shingle, with further information at https://www.shingleproject. org. This is accompanied by a manual, a suite of example domain discretisation BRML descriptions and the verification test engine presented in section 7.

All components of the Shingle package which have been under continued development since 2011 are free software, being released under the GNU General Public License version 3.0. Full details of the license, including the compatible copyright notices of third party routines included in the package, are included in the COPYING file in the source distribution.

**Acknowledgements**

The authors wish to acknowledge support from the Netherlands Organisation for Scientific Research (NWO, grant number 858.14.061), and also thank Gerben de Boer for discussions on OPeNDAP and its adoption within the Netherlands and more generally by the wider scientific community.

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
