# Peer review of "Shingle 2.0: generalising self-consistent and automated domain discretisation for multi-scale geophysical models $\dagger$"

_Geoscientific Model Development, 2017_

## Referee Comment (RC1) · Anonymous Referee #1 · 23 Jun 2017

This manuscript presents the general structure and design of the Shingle 2.0 library. The goals of the library are to allow the full description of domain discretizations in a reproducible and shareable manner. From this perspective meshes are an integral part of the overall model description. This contrasts with the somewhat ad-hoc manner in which meshing is often treated in today's literature. Shingle 2.0 uses the Spud library, which allows common model features to be exposed to users through a hierarchical options interface, diamond, that is easily extensible when new features are required.

The general idea of this library is excellent. Meshes and domain discretizations should be treated much better than they often currently are and allowing users to share and

build on other authors' work in a reproducible manner will certainly be helpful. I am however concerned that while this paper does a reasonable job of the difficult task of presenting the library, much of the theory (and the original version of the Shingle library) appears to be in a paper (Candy, A.S., 2016. A consistent approach to unstructured mesh generation for geophysical models.) that is still under review. This manuscript relies heavily on this paper, frequently citing and referencing it, and the authors have made it available online, which is useful, but it would seem odd if this manuscript was published first.

Beyond this manuscript the library appears to be well documented and I was able to install it however the claim is made (e.g. line 541) that deviations in the mesh are only expected to depend on the version of the shingle library. This seems like quite a bold claim, given that the library has a number of dependencies. These dependencies should be discussed in the manuscript - some are mentioned throughout but some more discussion or a table would be useful (a full list is provided in the manual).

A number of example snapshots are given but these are mostly taken from the afore-mentioned paper, Candy 2016. I think it would be very useful if a full worked example was included in this manuscript. This would demonstrate the workflow and could be used to direct potential users to more complete examples in the manual.

A worked example may also help to illuminate Figure 2, which is referenced a lot but did not help me to understand the manuscript very much. It's quite a confusing list set of arrows and labels, with no clear workflow presented. I realize there may be multiple possible workflows depending on how the user interacts with the library but these could be described much better in worked examples.

Some more minor technical issues:

- line 32: missing "is": "... - is likely to grow."

- line 49: first reference to table 1 (page 3) but then table 1 doesn't appear until page

9. Please move up.

- line 100: the sentence beginning "Its modular library framework, ..." is very long and unwieldy. Please break up.

- line 133: typo? "Lower-lever" -> "Lower-level"?

- line 148: Another unwieldy sentence. Consider changes marked by *: "The LibShingle library*,* central to the generalised approach (illustrated in figure 2)*,* is detailed in section 5 *and* ways to *interact* with the framework *are* presented in section 6. Examples and validation *are* covered in section 7,..."

- line 162: outcome*s*

- after equations 7 and 8: "identification elements" are not defined

- figure 4: make bigger (text width?) and higher resolution?

- line 313: "This information can *be* presented..."

- line 537: "... if possible, *is* better handled automatically..."

- line 673: "... in *the* COPYING *file*..."

---

## Referee Comment (RC2) · Anonymous Referee #2 · 27 Jun 2017

This paper describes Shingle 2.0 — a Python-based library for the manipulation of spatial domains and unstructured grids for geophysical problems. Through the use of a new XML-based file-format (BRML) and a hierarchy of publicly available software components, Shingle aims to standardise the process of managing the spatial constraints and unstructured grids associated with geophysical domains. To this end, a set of nine "tenets" for geophysical grid-generation are proposed, designed to facilitate the development of consistent and shareable frameworks for unstructured geophysical data and meshes.

The overall idea behind the Shingle library — the development of standardised approaches and formats for unstructured geophysical data — is interesting, as current methodologies are clearly ad-hoc. I do however have concerns regarding the distinction between the functionality of the Shingle package itself and the underlying libraries on which it depends. I suggest that a clear summary of the various dependencies be presented early in the paper, with an explicit delineation of functionality. Currently, it appears that:

- The Gmsh package provides the actual meshing capabilities, based on a geometry definition created by Shingle.

- The Spud package is used to support the XML-based BRML file-format. It's Diamond viewer is used for GUI-based file editing.

- Various packages (GDAL, shapely, pyproj) are used to support geometrical operations and queries.

- The pydap package is used for remote data access.

Does Shingle incorporate original algorithms and/or data processing facilities beyond those provided by the underlying libraries? If so, I suggest that these features be documented and novelty demonstrated, etc.

Additionally, I feel that the use of the Gmsh library should not be understated. While Shingle aims to overcome challenges related to the specification of the domain, geometric constraints, etc, I suggest that it is the underlying 'mesh-generation' process that is somewhat more algorithmically and computationally demanding.

Gmsh has also been used for geophysical grid-generation in the past (e.g. Lambrechts et al., 2008: "Multiscale mesh generation on the sphere"), along with a number of other algorithms and libraries, including: Jacobsen et al., 2013: "Parallel algorithms for planar and spherical Delaunay construction with an application to centroidal Voronoi

tessellations", Conroy et al., 2012: "ADMESH: An advanced, automatic unstructured mesh generator for shallow water models" and Holleman et al., 2013: (Stomel, in) "Numerical diffusion for flow-aligned unstructured grids with application to estuarine modeling", amongst others. I suggest including a brief review of these previous efforts, demonstrating the benefits of Shingle compared to existing alternatives.

The Candy, 2016 pre-print is referred to throughout, often to provide specific examples of functionality. I suggest that any examples referred to be included in the current paper directly. There appears to be some overlap between these papers, though the Candy, 2016 work appears to focus on more theoretical issues.

Page 2, line 32: ... [is] likely to grow.

Page 4, line 57: "... the meshing process is broken up over multiple parallel threads (as demonstrated in Candy, 2016), ..."

Does Shingle itself manage the parallel meshing process, or is this handled by the Gmsh library?

Page 5, line 102: develop[er]s

---

## Author Response (AR1)

**Delft University of Technology**

Environmental Fluid Mechanics Section
Faculty of Civil Engineering and Geosciences

Stevinweg 1
2628 CN Delft
The Netherlands

Email:          a.s.candy@tudelft.nl
Telephone:    +31 15 278 99 74
                    +44 775 2389 085

**Dr Adam Candy**  PhD (Imperial) MMath (Cantab)

Dr Dan Goldberg
Topical editor
Journal of Geoscientific Model Development

Tuesday 26 September 2017

Dear Dr Dan Goldberg,

**Re: gmd-2017-47 response to reviewers' comments**

I am writing with regard to the original research article titled, 'Shingle 2.0: generalising self-consistent and automated domain discretisation for multi-scale geophysical models', reference gmd-2017-47.

On behalf of my coauthor Julie Pietrzak and myself, we would like to thank the reviewers for their constructive comments which have enabled us to produce a revised version which we feel is significantly improved. We have considered each of the points in detail and our changes and response are included in the attached documents.

We would be grateful if you would consider our reply and the revised version of the paper, which has been uploaded separately, for publication with Geoscientific Model Development. There is ongoing interest in the approach and we are looking forward to publishing the research for the wider scientific community.

Thank you for your time considering the submission and we look forward to hearing from you.

Yours sincerely,

Dr Adam Candy

Encl.  Full response to reviews (46 pages), including
           Comments from referees (2 pages),
           Author's response (6 pages),
           Author's changes in the manuscript (2 pages),
           Marked-up version of the manuscript highlighting the changes made (36 pages).
        Additionally uploaded to accompany this response:
           Updated version of the manuscript (36 pages).

**Response to referees and manuscript update for gmd-2017-47:**
**Shingle 2.0: generalising self-consistent and automated domain discretisation for multi-scale geophysical models**

Adam S. Candy and Julie D. Pietrzak

September 28, 2017

**Contents**

Please note that references to lines in the manuscript, made in sections 2 and 3, have been updated to the new version of the manuscript with changes highlighted that is included in section 4. This version is largely the same as the annotated version submitted with the individual responses to the two reviews online, except that figure 4 has been redrawn to further improve legibility (see point 12 in section 3 for details). There are additionally a few further small changes, all of which are listed in section 3 and highlighted in section 4.

**1 Comments from Referees**

**1.1 Anonymous Referee #1**

**General comments**

This manuscript presents the general structure and design of the Shingle 2.0 library. The goals of the library are to allow the full description of domain discretizations in a reproducible and shareable manner. From this perspective meshes are an integral part of the overall model description. This contrasts with the somewhat ad-hoc manner in which meshing is often treated in today's literature. Shingle 2.0 uses the Spud library, which allows common model features to be exposed to users through a hierarchical options interface, diamond, that is easily extensible when new features are required.

The general idea of this library is excellent. Meshes and domain discretizations should be treated much better than they often currently are and allowing users to share and build on other authors' work in

a reproducible manner will certainly be helpful. I am however concerned that while this paper does a reasonable job of the difficult task of presenting the library, much of the theory (and the original version of the Shingle library) appears to be in a paper (Candy, A.S., 2016. A consistent approach to unstructured mesh generation for geophysical models.) that is still under review. This manuscript relies heavily on this paper, frequently citing and referencing it, and the authors have made it available online, which is useful, but it would seem odd if this manuscript was published first.

Beyond this manuscript the library appears to be well documented and I was able to install it however the claim is made (e.g. line 541) that deviations in the mesh are only expected to depend on the version of the shingle library. This seems like quite a bold claim, given that the library has a number of dependencies. These dependencies should be discussed in the manuscript - some are mentioned throughout but some more discussion or a table would be useful (a full list is provided in the manual).

A number of example snapshots are given but these are mostly taken from the aforementioned paper, Candy 2016. I think it would be very useful if a full worked example was included in this manuscript. This would demonstrate the workflow and could be used to direct potential users to more complete examples in the manual.

A worked example may also help to illuminate Figure 2, which is referenced a lot but did not help me to understand the manuscript very much. It's quite a confusing list set of arrows and labels, with no clear workflow presented. I realize there may be multiple possible workflows depending on how the user interacts with the library but these could be described much better in worked examples.

**Technical corrections**

– line 32: missing "is": "... - is likely to grow."

– line 49: first reference to table 1 (page 3) but then table 1 doesn't appear until page 9. Please move up.

– line 100: the sentence beginning "Its modular library framework, ..." is very long and unwieldy. Please break up.

– line 133: typo? "Lower-lever" → "Lower-level"?

– line 148: Another unwieldy sentence. Consider changes marked by *: "The LibShingle library*,* central to the generalised approach (illustrated in figure 2)*,* is detailed in section 5 *and* ways to *interact* with the framework *are* presented in section 6. Examples and validation *are* covered in section 7,..."

– line 162: outcome*s*

– after equations 7 and 8: "identification elements" are not defined

– figure 4: make bigger (text width?) and higher resolution?

– line 313: "This information can *be* presented..."

– line 537: "... if possible, *is* better handled automatically..."

– line 673: "... in *the* COPYING *file*..."

**1.2 Anonymous Referee #2**

**General comments**

This paper describes Shingle 2.0 a Python-based library for the manipulation of spatial domains and unstructured grids for geophysical problems. Through the use of a new XML-based file-format (BRML) and a hierarchy of publicly available software components, Shingle aims to standardise the process of managing the spatial constraints and unstructured grids associated with geophysical domains. To this end, a set of nine "tenets" for geophysical grid-generation are proposed, designed to facilitate the development of consistent and shareable frameworks for unstructured geophysical data and meshes.

The overall idea behind the Shingle library the development of standardised approaches and formats for unstructured geophysical data is interesting, as current methodologies are clearly ad-hoc. I do however have concerns regarding the distinction between the functionality of the Shingle package itself and the underlying libraries on which it depends. I suggest that a clear summary of the various dependencies be presented early in the paper, with an explicit delineation of functionality. Currently, it appears that:

– The Gmsh package provides the actual meshing capabilities, based on a geometry definition created by Shingle.

– The Spud package is used to support the XML-based BRML file-format. It's Diamond viewer is used for GUI-based file editing.

– Various packages (GDAL, shapely, pyproj) are used to support geometrical operations and queries.

– The pydap package is used for remote data access.

Does Shingle incorporate original algorithms and/or data processing facilities beyond those provided by the underlying libraries? If so, I suggest that these features be documented and novelty demonstrated, etc.

Gmsh has also been used for geophysical grid-generation in the past (e.g. Lambrechts et al., 2008: "Multiscale mesh generation on the sphere"), along with a number of other algorithms and libraries, including: Jacobsen et al., 2013: "Parallel algorithms for planar and spherical Delaunay construction with an application to centroidal Voronoi tessellations", Conroy et al., 2012: "ADMESH: An advanced, automatic unstructured mesh generator for shallow water models" and Holleman et al., 2013: (Stomel, in) "Numerical diffusion for flow-aligned unstructured grids with application to estuarine modeling", amongst others. I suggest including a brief review of these previous efforts, demonstrating the benefits of Shingle compared to existing alternatives.

Additionally, I feel that the use of the Gmsh library should not be understated. While Shingle aims to overcome challenges related to the specification of the domain, geometric constraints, etc, I suggest that it is the underlying 'mesh-generation' process that is somewhat more algorithmically and computationally demanding.

The Candy, 2016 pre-print is referred to throughout, often to provide specific examples of functionality. I suggest that any examples referred to be included in the current paper directly. There appears to be some overlap between these papers, though the Candy, 2016 work appears to focus on more theoretical issues.

**Technical corrections**

– Page 2, line 32: ... [is] likely to grow.

– Page 4, line 57: "... the meshing process is broken up over multiple parallel threads (as demonstrated in Candy, 2016), ..." Does Shingle manage the parallel meshing process, or is this handled by the Gmsh library?

– Page 5, line 102: develop[er]s

**2 Author's response**

**2.1 Anonymous Referee #1**

We would like to thank the reviewer for their constructive comments which have enabled us to produce a revised version which we feel is significantly improved. Parts of the review that have been included below are shown in *italics*. The changes made to the manuscript are highlighted in section 4.

**General comments**

*This manuscript presents the general structure and design of the Shingle 2.0 library. The goals of the library are to allow the full description of domain discretizations in a reproducible and shareable manner. From this perspective meshes are an integral part of the overall model description. This contrasts with the somewhat ad-hoc manner in which meshing is often treated in today's literature. Shingle 2.0 uses the Spud library, which allows common model features to be exposed to users through a hierarchical options interface, diamond, that is easily extensible when new features are required.*

*The general idea of this library is excellent. Meshes and domain discretizations should be treated much better than they often currently are and allowing users to share and build on other authors' work in a reproducible manner will certainly be helpful. I am however concerned that while this paper does a reasonable job of the difficult task of presenting the library, much of the theory (and the original version of the Shingle library) appears to be in a paper (Candy, A.S., 2016. A consistent approach to unstructured mesh generation for geophysical models.) that is still under review. This manuscript relies heavily on this paper, frequently citing and referencing it, and the authors have made it available online, which is useful, but it would seem odd if this manuscript was published first.*

We appreciate the comments on the use of the library. We also agree that there is a clear split in focus, that theory and a consistent approach is the main driver of Candy (2016) and handling very complex, multi-scale spatial discretisations with many constraints the motivation of this work. We have tried to ensure the distinct aims are listed and made clear in the outset of both. Preprints are openly available online. While both are under review we hope they will be published in a similar timeframe. This will open the overall approach to a wider audience in general.

*Beyond this manuscript the library appears to be well documented and I was able to install it however the claim is made (e.g. line 541) that deviations in the mesh are only expected to depend on the version of the shingle library. This seems like quite a bold claim, given that the library has a number of dependencies. These dependencies should be discussed in the manuscript - some are mentioned throughout but some more discussion or a table would be useful (a full list is provided in the manual).*

The reviewer is correct that dependencies can cause deviations in the output spatial discretisations. In fact there are a number of dependencies highlighted in line 541 (now lines 617–621). We agree this is a useful point to discuss further and have expanded this part of the paper in response. In order to clarify, the new table 3 has an exhaustive list of dependencies, together with the potential deviations they may cause, the risk and mitigation approaches employed. Line 541 (now 617–621) has also been modified to mention tessellation algorithm implementations and link to the table for more details.

In the large part this is an issue with all numerical simulation models which are linked to and use other libraries. The version of these depend on the build environment, which can vary between systems and over time. Here we are taking the opportunity to be more explicit about these dependencies. We thank the reviewer for the encouragement to clearly highlight the dependencies and their potential impact on the use of the library.

In addition to the dependencies that have the potential to cause deviations, we note that the new table 2, listing the functions in Shingle that use external libraries has also been added, supplementing the details in the schematic of figure 2.

> *A number of example snapshots are given but these are mostly taken from the aforementioned paper, Candy 2016. I think it would be very useful if a full worked example was included in this manuscript. This would demonstrate the workflow and could be used to direct potential users to more complete examples in the manual.*
>
> *A worked example may also help to illuminate Figure 2, which is referenced a lot but did not help me to understand the manuscript very much. It's quite a confusing list set of arrows and labels, with no clear workflow presented. I realize there may be multiple possible workflows depending on how the user interacts with the library but these could be described much better in worked examples.*

The 2010 Chile earthquake example was intended to be a full worked example in the paper and guide the reader through a sample workflow. A description of the domain is presented in words in (*) on page 4 from line 69. This is then shown entered into the Diamond GUI in figure 4, which in the background negotiates with the Shingle library to ensure options are valid and follow the structure shown in figure 3. This generates the BRML file (in XML) in figure 5. This is then processed by the Shingle library to generate the output spatial discretisation shown in figure 7.

The reviewer has highlighted that this was not made clear enough in the paper. In response we have added two sentences (lines 65–8) to explain the Chile 2010 case is used as a worked example, starting from line 65, straight into (*) and concluding with figure 7. To additionally help illuminate figure 2, this new text points out this follows the simplest high-level workflow illustrated across the top, from Diamond GUI to Shingle to mesh.

A selection of example discretisations are shown in figure 1, some of which appear in the paper Candy (2016). These are included to motivate the aims of the paper – that a generalised approach is needed, that is model-independent and applicable to a range of Earth Systems.

The only other case also appearing in Candy (2016) is part of figure 9, which was useful to include to highlight that global domains can be considered. The rest of figure 9 contains new studies on selected regions.

All other cases are new and do not appear elsewhere. This includes the full worked example of the 2010 Chile tsunami in figures 4, 5 and 7; the Caribbean Sea basin in figure 8; and new studies on selected regions of Antarctica in figure 9.

In addition to the main full worked example of the 2010 Chile tsunami that follows the simple Diamond to Shingle to mesh workflow, other ways to interact and full workflows are provided. Figure 6 includes three example Python codes, which with Shingle installed, directly run to give output spatial discretisations. The conclusions discuss Jupyter notebooks, which are a good interactive way to see and explore full workflows. Example notebooks are provided with the Shingle library (now noted in footnote 10, below line 700) and discussed in more detail in the manual.

**Technical corrections**

> *– line 32: missing "is": "… - is likely to grow."*

The word 'is' added such that the sentence reads better, as suggested.

– *line 49: first reference to table 1 (page 3) but then table 1 doesn't appear until page 9. Please move up.*

Table 1 has been moved up to appear directly following its first reference.

– *line 100: the sentence beginning "Its modular library framework, ..." is very long and unwieldy. Please break up.*

This sentence has been broken into two:

It has a modular library framework, with for example, geospatial operations, homeomorphic projections, meshing algorithms and model format writers the focus of distinct modular parts.

This together with the use of standard external libraries where possible allows development to remain in small sections of the code base such that developers can stay within their specialisms.

– *line 133: typo? "Lower-lever" → "Lower-level"?*

Corrected to 'Lower-level'.

– *line 148: Another unwieldy sentence. Consider changes marked by *: "The LibShingle library*,* central to the generalised approach (illustrated in figure 2)*,* is detailed in section 5 *and* ways to *interact* with the framework *are* presented in section 6. Examples and validation *are* covered in section 7,..."*

Sentence improved following suggestions.

– *line 162: outcome*s**

Corrected to 'outcomes'.

– *after equations 7 and 8: "identification elements" are not defined*

The text following equations 7 and 8 has been expanded to clarify the "identification elements", to become:

to give the full domain discretisation of $\Omega \subset \mathbb{R}^3$, consisting of a tessellation or honeycomb together with identification of the boundary and internal regions (i.e. $n_{\Gamma'}$ and $n_{\Omega'}$).

– *figure 4: make bigger (text width?) and higher resolution?*

Figure 4 has been regenerated at a higher resolution and widened to the text width as suggested.

– *line 313: "This information can *be* presented..."*

Sentence corrected to include 'be'.

– *line 537: "... if possible, *is* better handled automatically..."*

Sentence corrected to include 'is'.

– *line 673: "... in *the* COPYING *file*..."*

Sentence corrected following this suggestion.

**2.2   Anonymous Referee #2**

We would like to thank the reviewer for their constructive comments which have enabled us to produce a revised version which we feel is significantly improved. Parts of the review that have been included below are shown in *italics*. The changes made to the manuscript are highlighted in section 4.

**General comments**

*This paper describes Shingle 2.0 a Python-based library for the manipulation of spatial domains and unstructured grids for geophysical problems. Through the use of a new XML-based file-format (BRML) and a hierarchy of publicly available software components, Shingle aims to standardise the process of managing the spatial constraints and unstructured grids associated with geophysical domains. To this end, a set of nine "tenets" for geophysical grid-generation are proposed, designed to facilitate the development of consistent and shareable frameworks for unstructured geophysical data and meshes.*

*The overall idea behind the Shingle library the development of standardised approaches and formats for unstructured geophysical data is interesting, as current methodologies are clearly ad-hoc. I do however have concerns regarding the distinction between the functionality of the Shingle package itself and the underlying libraries on which it depends. I suggest that a clear summary of the various dependencies be presented early in the paper, with an explicit delineation of functionality. Currently, it appears that:*

*– The Gmsh package provides the actual meshing capabilities, based on a geometry definition created by Shingle.*

*– The Spud package is used to support the XML-based BRML file-format. It's Diamond viewer is used for GUI-based file editing.*

*– Various packages (GDAL, shapely, pyproj) are used to support geometrical operations and queries.*

*– The pydap package is used for remote data access.*

*Does Shingle incorporate original algorithms and/or data processing facilities beyond those provided by the underlying libraries? If so, I suggest that these features be documented and novelty demonstrated, etc.*

We agree that it is helpful to include a list of the dependencies and discuss their function, to supplement the details in the section *'5.1 Built on standard libraries'* of *'5 LibShingle, the Shingle library framework'*. We have added the new table 2 there which lists the functions of Shingle that depend on external libraries. This acts as a good summary, including the details the reviewer suggested above and expanding where possible. Notably this includes reference to the Gmsh library for tessellation algorithms, Spud and Diamond for parameter management and GUI, standard libraries for geospatial operations such as GDAL and OPeNDAP libraries for remote data access.

We also took the opportunity to review the text on how dependencies may affect the reproducibility of output that appears in the conclusions in lines 617–621. This covers both internal and external factors, including these external libraries. The new table 3 lists these dependencies, together with the potential deviations they may cause, the risk and mitigation approaches employed. In the large part this is an issue with all numerical simulation models which are linked to and use other libraries. The versions of these depend on the build environment, which can vary between systems and over time. Here we are taking the opportunity to be more explicit about these dependencies.

We see the use of external libraries a strength of the approach (cf. tenet 9). This is particularly the case where these are standard, well-regarded and well-tested. These are supported by the community, undergo strict verification testing and validated in a wide range of applications. This is the strength of joint, community efforts such as the PETSc (Portable, Extensible Toolkit for Scientific Computation, `https://www.mcs.anl.gov/petsc`) library of numerical algorithms.

In some cases it may be easier to implement an algorithm from scratch, but this would yield yet another ad hoc approach, more code and features to manage directly. Moreover, this will not benefit from future new features and support added to external libraries. These focused efforts arguably do a better job in the long term and provide a more sustainable approach than a reimplementation or development from scratch.

*Gmsh has also been used for geophysical grid-generation in the past (e.g. Lambrechts et al., 2008: "Multiscale mesh generation on the sphere"), along with a number of other algorithms and libraries, including: Jacobsen et al., 2013: "Parallel algorithms for planar and spherical Delaunay construction with an application to centroidal*

*Voronoi tessellations", Conroy et al., 2012: "ADMESH: An advanced, automatic unstructured mesh generator for shallow water models" and Holleman et al., 2013: (Stomel, in) "Numerical diffusion for flow-aligned unstructured grids with application to estuarine modeling", amongst others. I suggest including a brief review of these previous efforts, demonstrating the benefits of Shingle compared to existing alternatives.*

We agree that the paper would benefit from a review of previous efforts. Moreover, whilst the advantages of the approach are expressed throughout the paper, we agree it is also helpful to collect and focus points here discussing the advantages of Shingle compared to existing alternatives. In response we have added the new section 2.4 on *'Tessellation algorithms and existing grid generation approaches'*, that we think significantly improves the manuscript on this point.

*Additionally, I feel that the use of the Gmsh library should not be understated. While Shingle aims to overcome challenges related to the specification of the domain, geometric constraints, etc, I suggest that it is the underlying 'mesh-generation' process that is somewhat more algorithmically and computationally demanding.*

We agree that the quality of a spatial discretisation is directly dependent on both (i) the algorithm used to create a tessellation and (ii) accuracy and self-consistency of constraints under which the former operates (now stated in section 2.4, lines 231–41). Here we seek to improve the latter with the nine tenets in mind, and as the reviewer highlights, overcome the challenges related to the specification of the domain and geometric constraints.

We strongly support the reviewer's point that Gmsh has an important role in the approach and for mesh generation in general. We have emphasized this throughout the paper, and notably in lines 200, 223, 235, 287, 408, 529, 609, the new section 2.4 and new tables 2 and 3. Line 200 of section 2.4 in particular states, *'The general-purpose three-dimensional meshing library Gmsh (Geuzaine and Remacle, 2009) has been used to make significant progress in ocean modelling on unstructured meshes (e.g. see Legrand et al., 2000; White et al., 2008; van Scheltinga et al., 2010; Gourgue et al., 2013; Thomas et al., 2014)'*. It was important to facilitate the use of other libraries (line 528) in line with the ninth tenet on standardisation. Whilst use of other libraries is possible, compared to many alternatives, Gmsh does a very good job at adhering to the constraints provided, is relatively robust and provides access to multiple tessellation algorithms through a common API, again in support of tenet 9 (all points now added to section 6 from line 531).

*The Candy, 2016 pre-print is referred to throughout, often to provide specific examples of functionality. I suggest that any examples referred to be included in the current paper directly. There appears to be some overlap between these papers, though the Candy, 2016 work appears to focus on more theoretical issues.*

We agree with the reviewer that the paper Candy (2016) has separate, distinct aims focusing on consistency and theoretical issues. The reviewer is correct that there is necessarily some overlap. We believe this is needed, to draw connections between the works and motivate aims here, but have tried to minimise as much as possible.

A selection of example discretisations are shown in figure 1, some of which appear in the paper Candy (2016). These are included to motivate the aims of the paper – that a generalised approach is needed that is model-independent and applicable to a range of Earth Systems. The only other case also appearing in Candy (2016) is part of figure 9, which was useful to include to highlight that global domains can be considered. All other cases are new and do not appear elsewhere. This includes the full worked example of the 2010 Chile tsunami in figures 4, 5 and 7; the Caribbean Sea basin in figure 8; and new studies on selected regions of Antarctica also in figure 9.

The reviewer is correct that Candy (2016) is referenced throughout to connect the efforts. We have reviewed all of these points in the text and where appropriate referenced material is included or summarised. Notably the key results of the constraints and tenets are included in a reduced form in section

2.1 and table 1, respectively. The global case is included in figure 9 as the reviewer suggests. The only other reference back to examples is made in the sentence in lines 138–41, which is not to provide specific examples of functionality, but a broad statement is made to highlight the generalised approach and range of cases this is applicable to. We were careful to ensure this work included distinct example cases, with the 2010 Chile tsunami domain acting as the main full worked example.

**Technical corrections**

– *Page 2, line 32: … [is] likely to grow.*

The word 'is' added such that the sentence reads better, as suggested.

– *Page 4, line 57: "… the meshing process is broken up over multiple parallel threads (as demonstrated in Candy, 2016), …" Does Shingle itself manage the parallel meshing process, or is this handled by the Gmsh library?*

Yes, the Shingle library handles the discretisation over multiple parallel threads. Gmsh is currently used in serial. This not a limitation of the Shingle library. We have recently seen a paper published demonstrating Gmsh running over multiple threads at a fine grain level. This would certainly benefit the approach here and we look forward to using Gmsh in this way, parallel over multiple threads, in the future.

– *Page 5, line 102: develop[er]s*

Corrected to 'developers'.

**4   Marked-up version of the manuscript highlighting the changes made**

A copy of the manuscript with changes highlighted follows. New, added content is marked with a green wavy underscore, whilst removed material appears in orange with a strikethrough.

[revised manuscript text omitted]